# MINIMAX OPTIMAL ADVERSARIAL REINFORCEMENT LEARNING

**Yudan Wang[1], Kaiyi Ji[2], Ming Shi[3] & Shaofeng Zou[1]**
[1]Electrical, Computer and Energy Engineering, Arizona State University
[2]Computer Science and Engineering, University at Buffalo
[3]Electrical Engineering, University at Buffalo
{ywan1645, zou}@asu.edu; {kaiyiji, mshi24}@buffalo.edu

## ABSTRACT

Consider episodic Markov decision processes (MDPs) with adversarially chosen transition kernels, where the transition kernel is adversarially chosen at each episode. Prior works have established regret upper bounds of $\widetilde{\mathcal{O}}(\sqrt{T} + C^P)$, where $T$ is the number of episodes and $C^P$ quantifies the degree of adversarial change in the transition dynamics. This regret bound may scale as large as $\mathcal{O}(T)$, leading to a linear regret. This raises a fundamental question: *Can sublinear regret be achieved under fully adversarial transition kernels?* We answer this question affirmatively. First, we show that the optimal policy for MDPs with adversarial transition kernels must be history-dependent. We then design an algorithm of Adversarial Dynamics Follow-the-Regularized-Leader (AD-FTRL), and prove that it achieves a sublinear regret of $\widetilde{\mathcal{O}}(\sqrt{(|\mathcal{S}||\mathcal{A}|)^K T})$, where $K$ is the horizon length, $|\mathcal{S}|$ is the number of states, and $|\mathcal{A}|$ is the number of actions. Such a regret cannot be achieved by simply solving this problem as a contextual bandit. We further construct a hard MDP instance and prove a matching lower bound on the regret, which thereby demonstrates the *minimax optimality* of our algorithm.

## 1 INTRODUCTION

Reinforcement learning (RL) (Sutton et al., 1998) is a general framework for sequential decision-making, where a learner interacts with an unknown environment in order to learn the optimal policy over time. Consider the episodic setting as an example, where the number of interactions between the agent and the environment is a fixed number $K$ in each episode. At each stage $k \in \{0, ..., K-1\}$, the learner takes action $a^k$ according to the current state $s^k$ or histories $h^k = \{s^0, a^0, ..., s^{k-1}, a^{k-1}, s^k\}$. Then, the learner observe the next state $s^{k+1}$ which is sampled from a unknown transition kernel $P(\cdot|s^k, a^k)$, and received a loss $\ell(s^k, a^k)$. The interaction terminates at the step $K$, then a new episode starts. The goal of RL is to find a policy sequence to minimize the regret, which is defined as the gap between the total loss obtained during learning and under a fixed optimal policy.

Existing works mostly focus on MDPs with fixed transition kernel and loss function, which do not change across episodes (Sutton et al., 1998; Auer et al., 2008; Azar et al., 2017; Jin et al., 2020b). However, in practice, the environment can be time-varying or subject to adversarial corruptions. Recent studies formulate this problem as adversarial MDPs, where loss and/or transition kernels may be chosen adversarially at each episode. One line of research focuses on the case with adversarial loss, where the loss function is adversarially perturbed but the transition kernel is the same across episodes (Even-Dar et al., 2009; Neu et al., 2010; Zimin & Neu, 2013; Jin et al., 2020a; 2021; Rosenberg & Mansour, 2019). When the transition kernel is also adversarially chosen at each episode, the problem becomes significantly more challenging, and studies on this problem are quite limited. For this problem, recent works (Jin et al., 2023; Wei et al., 2022) showed regret bounds that scale with a term of $C^P$, which quantifies the corruption level of the transition kernels. Notably, in the fully adversarial case, $C^P$ can be as large as $\mathcal{O}(T)$, leading to a linear regret. This raises a fundamental question:

> *Can we develop an algorithm for RL with adversarially chosen transition kernels, and prove a sublinear regret?*

In this paper, we answer this question affirmatively. Specifically, we develop a learning algorithm, Adversarial Dynamics Follow-the-Regularized-Leader (AD-FTRL), that operates under *bandit feedback* with unknown adversarially chosen transition kernels. We show that it achieves a sublinear regret bound of $\widetilde{\mathcal{O}}(\sqrt{(|\mathcal{S}||\mathcal{A}|)^K T})$, where $K$ is the horizon length, $|\mathcal{S}|$ is the number of states, and $|\mathcal{A}|$ is the number of actions. Furthermore, we demonstrate that this regret bound is minimax optimal by constructing a matching lower bound, thereby establishing the minimax optimality results for MDPs with adversarially chosen transition kernels.

## 1.1 CHALLENGES

In standard episodic MDPs with a fixed transition kernel, it is proved that there exists a Markov policy that is optimal (Sutton et al., 1998). However, the same may not be true for adversarial MDPs, though some existing works (Jin et al., 2021; 2023) consider Markov policies for simplicity of analysis. A simple example can be constructed as follows. Consider the case where the initial state reveals the transition kernel chosen by the adversary; then the optimal policy must be history dependent (i.e., it depends on the initial state). Therefore, we need to search over the more expressive class of history-dependent policies, which is more challenging to tackle.

Designing algorithms for history-dependent policies is substantially more difficult than Markov policies, noting that the history space can be significantly larger than the state space. Moreover, the learner operates with bandit feedback and lacks knowledge of the transition kernels $\{P_t\}_{t \in [T]}$, which are chosen adversarially and may vary arbitrarily across episodes. Estimating these transitions is infeasible, and even estimating their average $\bar{P}$ is insufficient, as the optimal policy for $\bar{P}$ may be far from optimal for the sequence $\{P_t\}$. This mismatch introduces an unavoidable regret penalty, quantified by the $C^P$ term in existing analyses (Jin et al., 2023; Wei et al., 2022; Chen et al., 2021; Lykouris et al., 2021; Wei et al., 2022). To address these challenges, we avoid estimating $\bar{P}$ altogether and instead propose an approach based on importance sampling and trajectory-level occupancy measures (a type of visitation measure). Furthermore, we carefully design a regularization term to ensure a sublinear regret bound, as the occupancy measures are affected by the time-varying transitions—a complication that does not arise in settings with adversarial losses but fixed transition kernels (including fixed kernel with corruption).

However, we notice that our bound is still exponential in $K$, which is less favorable. This also naturally raises another question: *Can the regret of $\sqrt{(|\mathcal{S}||\mathcal{A}|)^K}$ be improved?* To answer it and also understand the minimax optimality, we face two major technical challenges. First, in adversarial RL settings, the study of history-dependent policies is still scarce. As a result, the lack of structural understanding of such policies makes it particularly difficult to construct an appropriate *hard MDP instance* for minimax lower bound derivation. Second, deriving a tighter lower bound requires reducing the regret minimization problem to a composite hypothesis testing problem—a necessity imposed by the adversarial nature of the transition dynamics. This reduction forms the core of our technical contribution. However, unlike binary or multi-hypothesis testing scenarios commonly studied in standard RL, composite hypothesis testing and its corresponding regret analysis present significantly greater challenges, as is widely acknowledged.

## 1.2 CONTRIBUTIONS

1. *Characterization of the Optimal Policy.* We first prove that the optimal policy that minimizes cumulative loss with adversarially chosen transition kernel must be a history-dependent policy, instead of a Markov one.

2. *Algorithm Design for Adversarial Chosen Transition Kernels.* Under the challenging setting of *bandit feedback losses* and *unknown, adversarially chosen transition kernels*, we propose an Adversarial Dynamic Follow-the-Regularized-Leader (AD-FTRL) algorithm that updates a history-dependent policy. Moreover, with a carefully designed regularization term, we prove that our algorithm achieves a sublinear regret bound of $\widetilde{\mathcal{O}}(\sqrt{(|\mathcal{S}||\mathcal{A}|)^K T})$, without requiring prior knowledge of the transition kernels.

3. *Minimax Optimal Regret Bound.* We carefully design a *hard instance of MDPs* with regret lower bound of $\Omega(\sqrt{(|\mathcal{S}||\mathcal{A}|)^K T})$ order, matching the regret upper bound of the AD-FTRL algorithm, which shows the minimax optimality of our results. Compared to the lower bound presented in (Tian et al., 2021), our result is tighter and explicitly shows that the

regret scales with both the state and action space dimensions. This optimal minimax regret bound confirms the fundamental difficulty of the problem and the minimax optimality of our algorithm. In our proof, we introduce a new analytical approach for handling adversarial or time-varying transitions using information-theoretic tools from composite hypothesis testing. Our newly constructed *hard instance of MDPs* and accompanying analysis framework provide a unified and complete solution for the minimax optimal regret bound of adversarial RL.

## 1.3 RELATED WORKS

***Upper Bound Analysis.*** Firstly, we introduce the work for the regret against a fixed optimal policy among the total episodes. For the case with adversarial loss functions but a fixed transition kernel, adversarial RL has been widely studied in previous works (Even-Dar et al., 2009; Zimin & Neu, 2013; Neu et al., 2010; Dick et al., 2014; Jin & Luo, 2020; Rosenberg & Mansour, 2019; Jin et al., 2020a; 2021; Rosenberg & Mansour, 2019; Chen & Luo, 2021; Luo et al., 2021; Dann et al., 2023a;b). Among these works, with bandit feedback of adversarially chosen losses and fixed but unknown transition kernel, the works (Jin et al., 2020a; 2021) provide different algorithms obtaining a sublinear $\widetilde{\mathcal{O}}(\sqrt{|\mathcal{S}||\mathcal{A}|T})$ regret bound, which also holds in fully adversarial losses setting.

However, when the transition kernel at each episode is also adversarially chosen, the problem becomes challenging, and related studies are limited. The work (Abbasi Yadkori et al., 2013) provides an algorithm for adversarially chosen but known transition kernels. When the adversarially chosen transition kernels are unknown, previous works (Jin et al., 2023; Wei et al., 2022; Chen et al., 2021; Lykouris et al., 2021; Wei et al., 2022; Deng et al., 2024) estimate the central/true transition kernel and update the policy based on the estimated one. The best regret upper bound from these works is $\widetilde{\mathcal{O}}(\sqrt{|\mathcal{S}||\mathcal{A}|T} + C^P)$, which performs well when the corruption level is sublinear in $T$. However, under a fully adversarial setting, the corruption level becomes linear in $T$, i.e., $C^P = \mathcal{O}(T)$, leading to a linear regret bound. In contrast, our method directly estimates the visitation measure to minimize the regret, and reaches a sublinear $\widetilde{\mathcal{O}}(\sqrt{(|\mathcal{S}||\mathcal{A}|)^K T})$ regret.

Finally, we note another line of research on *non-stationary reinforcement learning*. These works also include different non-stationary measure terms in their dynamic regret bounds, such as the number of switches in the environment (Auer et al., 2008; Gajane et al., 2018), or variation/corruption measures (Wei & Luo, 2021; Cheung et al., 2023; Li et al., 2024b;a). However, most of them focus on dynamic regret, which evaluates the learner's performance relative to the optimal sequence of policies that may change over time, which are generally not directly comparable to ours.

Recent work has extended adversarial reinforcement learning beyond tabular settings. (Cai et al., 2020) analyzed adversarial rewards under linear function approximation, and (He et al., 2022) achieved near-optimal guarantees for adversarial linear mixture MDPs. More recently, (Ye et al., 2024) studied adversarial corruption of the transition kernel under general function approximation, obtaining near-optimal regret bounds. However, these results are not directly comparable to ours.

***Lower Bound Analysis.*** For standard RL with a fixed but unknown transition kernel and loss function, Auer et al. (2008) provides a regret lower bound under the average-reward setting. Besides, (Azar et al., 2017) provides a minimax optimal regret bound for finite-horizon RL problems. In the episodic RL setting, (Auer et al., 2008) claims a regret lower bound based on average-reward analysis, which is later improved by (Jin et al., 2018). However, neither work (Auer et al., 2008; Jin et al., 2018) provides a complete or rigorous proof of the episodic RL problem. Recently, (Domingues et al., 2021) offers unified and complete proofs for regret lower bounds, establishing that the regret must be at least $\Omega(\sqrt{|\mathcal{S}||\mathcal{A}|T})$ in the episodic RL setting.

Under bandit feedback with adversarial losses and an unknown fixed transition kernel, the regret lower bound for vanilla episodic RL is often stated as the lower bound for the adversarial loss setting, as in (Jin et al., 2020a). However, in the setting with adversarial transitions, the regret bound analysis becomes challenging. As a related setting, Markov game, a lower bound of $\Omega(\sqrt{2^K T})$ is derived in (Tian et al., 2021). However, it is unclear whether this bound is tight or achievable. Moreover, the bound does not specify the dependence on the size of the state space and/or action space. In our work, we present an algorithm and prove that the minimax-optimal regret reaches $\Omega(\sqrt{(|\mathcal{S}||\mathcal{A}|)^K T})$.

## 2 PRELIMINARIES

We consider the episodic setting, where a learner interacts with a sequence of $T$ adversarial episodic MDPs with time-varying transitions and losses, which can represented by the tuple $(\mathcal{S}, \mathcal{A}, K, \{P_t\}_{t\in[T]}, \{\ell_t\}_{t\in[T]})$, here $[T] = \{1, 2, ..., T\}$. All MDPs share the same joint state space $\mathcal{S}$ and joint action space $\mathcal{A}$. We adopt a layered MDP structure and assume, without loss of generality, that $\mathcal{S}$ is partitioned into $K + 1$ disjoint subsets $\mathcal{S}^0, \ldots, \mathcal{S}^K$, where $\mathcal{S}^K = \{s^K\}$ contains the terminal state and no actions are taken; $\mathcal{A}$ is partitioned into $K$ disjoint subsets $\mathcal{A}^0, \ldots, \mathcal{A}^{K-1}$. Transitions are allowed only between consecutive layers. We define the history space at stage $k \leq K$ as $\mathcal{H}^k := (\otimes_{j<k}\mathcal{S}^j \otimes_{j<k} \mathcal{A}^j) \times \mathcal{S}^k$. The environment selects the transition kernels $\{P_t\}_{t\in[T]}$ and loss functions $\{\ell_t\}_{t\in[T]}$ adversarially in advance, given the learner's algorithm. These sequences remain fixed and unknown to the learner. In each episode $t$, the learner executes a history-dependent policy $\pi_t := \otimes_{k=0}^{K-1}\pi_t^k$, where $\pi_t^k : \mathcal{H}^k \to \Delta(\mathcal{A})$. Unlike Markov policies, each $\pi_t^k$ maps full histories to actions. The learner begins at $s_t^0$, sets $h_t^0 = \{s_t^0\}$, and at each stage $k < K$, selects $a_t^k \sim \pi_t^k(\cdot \mid h_t^k)$, receives loss $\ell_t(s_t^k, a_t^k)$, and transitions to $s_t^{k+1} \sim P_t(\cdot \mid s_t^k, a_t^k)$, with updated history $h_t^{k+1} = h_t^k \cup \{a_t^k, \ell_t(s_t^k, a_t^k), s_t^{k+1}\}$.

Although the learner's policy is history-dependent, the environment's transition dynamics and loss functions are Markovian, consistent with prior works such as (Jin et al., 2023). The learner has no prior knowledge of these functions. After each episode, only the losses for visited state-action pairs are revealed (*bandit feedback losses*), and the transition kernels remain entirely hidden.

To simplify notation, we assume that each layer has a fixed number of states and actions: $|\mathcal{S}| := |\mathcal{S}^k|$, $|\mathcal{A}| := |\mathcal{A}^k|$ for all $k < K$. Additionally, we define $a^K = $ null and set the terminal loss as $\ell(s^K, a^K) := \ell(s^K)$. Then, let $\tau_t = \{s_t^0, a_t^0, \ldots, s_t^{K-1}, a_t^{K-1}, s_t^K\}$ denote the trajectory generated at episode $t$ under transition kernel $P_t$, loss function $\ell_t$, and policy $\pi$. Define the trajectory space as $\mathcal{C}_\tau = (\otimes_{j<K}\mathcal{S}_j \otimes \mathcal{A}_j) \otimes \mathcal{S}_K$, and the cumulative loss of a trajectory as $\ell_t(\tau) := \sum_{k=0}^{K} \ell_t(s^k, a^k)$. Given a history-dependent policy $\pi$, the value function is defined as $V_t(\pi) := V_{P_t}(\pi) = E\left[\ell_t(\tau_t) \mid P_t, \pi\right]$, which is the expected cumulative loss when executing $\pi$ in the $t$-th MDP. The regret against any policy $\pi$ is then $\text{Reg}_T(\pi) = \mathbb{E}\left[\sum_{t=1}^{T}(V_t(\pi_t) - V_t(\pi))\right]$. Let $\pi^*$ denote an optimal policy in the history-dependent policy class that maximizes this regret, i.e.,

$$\text{Reg}_T(\pi^*) = \max_\pi \text{Reg}_T(\pi).$$

For simplicity, we write $\text{Reg}_T := \text{Reg}_T(\pi^*)$ as shorthand. Therefore, the aim of the algorithm is to minimize the regret $\text{Reg}_T$ under *bandit feedback losses* and *unknown transition dynamics* setting.

## 3 WARM-UP: CHALLENGES AND SOLUTIONS UNDER ADVERSARIAL DYNAMICS

***History-Dependent Policy Class.*** In this paper, unlike prior work such as (Jin et al., 2023), we update policies within the history-dependent policy class rather than the Markov class. Below, we explain the motivation for this design choice. Generally, the Markov policy can be regarded as a special case of history-dependent policies. For standard episodic MDPs with fixed transitions and loss functions, (Sutton et al., 1998) proved that the optimal policy over the history-dependent class is Markovian. Consequently, many subsequent works optimize within the Markov policy set. In the adversarial loss setting with a fixed transition kernel, the problem can be reduced of learning under a sequence of loss functions $\{\ell_t\}_{t\in[T]}$ to optimizing against the average loss $\bar{\ell} = \frac{1}{T}\sum_{t\in[T]} \ell_t$, where optimal Markov policy is similar to the standard episodic MDP setting.

However, under adversarially varying transitions, existing studies remain unclear whether the optimal policy remains Markovian. We present a counterexample (Figure 1) showing that history-dependent policies can outperform Markov ones. Under the transition sequence $\{P_1, P_2, P_1, P_2, P_1\}$, let:

$$\pi^*_{\text{Markov}} = \underset{\pi \in \text{Markov policy class}}{\arg\min} \bar{V}(\pi), \qquad \pi^*_{\text{his}} = \underset{\pi \in \text{history-Dependent policy class}}{\arg\min} \bar{V}(\pi).$$

At stage 1, the optimal Markov policy sets $\pi^*_{\text{Markov}}(s^2 = \circ) = 0$ and achieves $\bar{V}(\pi^*_{\text{Markov}}) = \frac{2}{5}$. In contrast, a history-dependent policy that sets $\pi^*_{\text{his}}(s^0 = \circ, a^0 = 0, s^1 = \star, a^1 = 0, s^2 = \circ) = 0$ and $\pi^*_{\text{his}}(s^0 = \circ, a^0 = 0, s^1 = \circ, a^1 = 0, s^2 = \circ) = 1$ achieves $\bar{V}(\pi^*_{\text{his}}) = 0$.

This demonstrates that the Markov optimal policy can be strictly suboptimal, with a non-vanishing gap in expected return. Motivated by this, we restrict our analysis to history-dependent policies throughout the paper.

***Occupancy Measure.*** The *occupancy measure* is a widely used concept in adversarial RL, which quantifies the visitation frequency over the probability space (e.g., the state–action pairs). For a fixed transition kernel $P$ and stochastic policy $\pi$, the state-action occupancy measure $p_{P,\pi} : \mathcal{S} \times \mathcal{A} \to [0, 1]$ defines the probability of visiting each $(s, a)$ pair.

In adversarial reinforcement learning, the goal is to minimize the total regret: $\text{Reg}_T = \sum_{t=1}^{T} V_t(\pi_t) - V_t(\pi^*)$. To better illustrate the challenge in this setting, we consider a simplified adversarial setting where the loss function $\ell$ is fixed, but the transition kernels $\{P_t\}_{t=1}^{T}$ vary adversarially. In this case, the regret can be rewritten as: $\text{Reg}_T = \sum_{t=1}^{T} \langle p_{P_t,\pi_t} - p_{P_t,\pi^*}, \ell \rangle$. Minimizing this regret requires access to the state-action occupancy measure $p_{P_t,\pi}$ for policy $\pi$ and each transition kernel $P_t$. However, under adversarial and unknown transitions, estimating these quantities is infeasible: the learner only observes sampled

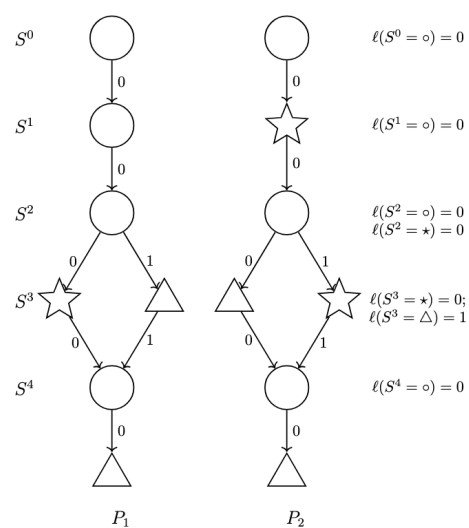

Figure 1: Counterexample: states $\circ, \star, \triangle$; actions $0, 1$; transitions alternate between $P_1$ and $P_2$.

state-action-next-state triplets $(s_t^k, a_t^k, s_t^{k+1})$ from interactions with the environment and cannot access the full transition dynamics. Instead, the learner can only estimate the average transition kernel $\bar{P} = \frac{1}{T} \sum_{t=1}^{T} P_t$ and use it to compute the corresponding occupancy $p_{\bar{P},\pi}$, which is often treated as a proxy for the true dynamics in prior works (Jin et al., 2023).

However, the average occupancy measure $\bar{p}_\pi = \frac{1}{T} \sum_{t=1}^{T} p_{P_t,\pi}$ is generally inaccurate and differs from the occupancy under the average transition kernel, i.e., $\bar{p}_\pi \neq p_{\bar{P},\pi}$. As a result, for any fixed policy $\pi$, there exists a gap introduced by the mismatch across $T$ episodes:

$$\sum_{t=1}^{T} V_{P_t}(\pi) - V_{\bar{P}}(\pi) = \sum_{t=1}^{T} \langle \bar{p}_\pi - p_{\bar{P},\pi}, \ell \rangle.$$

It further introduces an additional corruption term $\mathcal{O}(C^P)$ in regret bounds. On the other hand, the average occupancy measure $\bar{p}_\pi$ may not even correspond to any realizable transition model, further limiting the effectiveness of transition-based estimation approaches in adversarial settings.

To overcome this, we directly estimate the average trajectory occupancy measure, avoiding reliance on transition estimation. Define the trajectory-level occupancy $q_{P,\pi} : \mathcal{C}_\tau \to [0, 1]$, where for $\tau = (s^0, a^0, \ldots, s^K, a^K, s^{K+1})$: $q_{P,\pi}(\tau) = P(s^0) \prod_{k=0}^{K-1} \pi(a^k \mid h^k) \cdot P(s^{k+1} \mid s^k, a^k)$. Unlike prior methods that estimate state-action occupancy via known transitions, we operate directly in trajectory space. We estimate $q_{P_t,\pi}(\tau)$ using trajectories generated under a behavior policy $\pi_t$ via importance sampling. The following lemma provides an unbiased estimator:

**Lemma 3.1** (Trajectory Occupancy via Importance Sampling). *Let $\pi_t \in \mathcal{C}_{\pi,\epsilon}$ be the behavior policy at episode $t$, and let $\tau = (s^0, a^0, \ldots, s^K, a^K, s^{K+1}) \in \mathcal{C}_\tau$, where $\mathcal{C}_{\pi,\epsilon}$ is an $\epsilon$-greedy class (i.e., $\min_{k,h^k,a^k} \pi^k(a^k \mid h^k) \geq \epsilon$).*

*Then for any target policy $\pi$:*

$$\mathbb{E}_{\tau_t \sim q_{\pi_t, P_t}} \left[ \frac{\prod_{k=0}^{K-1} \pi(a_t^k \mid h_t^k)}{\prod_{k=0}^{K-1} \pi_t(a_t^k \mid h_t^k)} \cdot \mathbb{I}_{\tau_t = \tau} \right] = q_{\pi, P_t}(\tau).$$

This gives an unbiased estimator for any $\pi$, bypassing the need to estimate $P_t$.

Given trajectory $\tau_t \sim q_{\pi_t, P_t}$, we further construct:

$$\hat{q}_{\pi, P_t} = \left\{ \frac{\prod_{k=0}^{K-1} \pi(a_t^k \mid h_t^k)}{\prod_{k=0}^{K-1} \pi_t(a_t^k \mid h_t^k)} \cdot \mathbb{I}_{\tau_t = \tau} \right\}_{\tau \in \mathcal{C}_\tau}, \quad \hat{\ell}_t = \{\ell_t(\tau_t) \cdot \mathbb{I}_{\tau_t = \tau}\}_{\tau \in \mathcal{C}_\tau}.$$

The estimated return at episode $t$ can be written as: $\hat{V}_t(\pi) = \langle \hat{q}_{\pi, P_t}, \ell_t \rangle = \langle \hat{q}_{\pi, P_t}, \hat{\ell}_t \rangle$, which is unbiased: $\mathbb{E}[\hat{V}_t(\pi)] = V_t(\pi)$ and $\mathbb{E}[\hat{q}_{\pi, P_t}] = q_{\pi, P_t}$. Therefore, rather than estimating adversarial transitions, we can directly estimate trajectory occupancy using importance sampling. Next, we will show that this technique avoids the corruption penalty from estimating $\bar{P}$, enabling sublinear regret under adversarial dynamics and bandit feedback.

**Remark 1.** *A tempting idea to solve adversarial dynamic episodic MDPs is to reduce them to a bandit problem (considering a nontrivial horizon $H > 1$). However, in our setting, the transitions are **unknown** and they **cannot be estimated** accurately because the average occupancy measure differs from the occupancy measure under averaged transitions, i.e. $\bar{p}_\pi \neq p_{\bar{P}, \pi}$. Consequently, the standard mapping from an episodic MDP with known transitions to a bandit model does not apply. Another **strictly sub-optimal** approach is to treat each policy as an arm, thereby reducing the problem to a multi-armed bandit (MAB) setting. However, this leads to extremely large regret because it ignores the intrinsic structure and dependencies within the MDP. A remaining approach treats each length $H$ action sequence, $(a_0, \ldots, a_{K-1})$, as a single arm which yields a bandit with $|\mathcal{A}|^H$ arms. However, the best open-loop sequence is **strictly suboptimal** since it cannot exploit state feedback within the episode. More broadly, actions influence future state distributions, so the contexts observed **later depend on earlier** actions, which violates the contextual bandit assumption that the context law is exogenous. Together, these facts imply that adversarial dynamic episodic MDPs with unknown transitions cannot, in general, be solved by reducing them to bandit problems.*

## 4 ADVERSARIAL DYNAMICS FTRL ALGORITHM

Under adversarial dynamics, the Follow-the-Regularized-Leader (FTRL) framework is a well-established and powerful method for deriving online learning algorithms, particularly in settings where the environment changes over time. In our work, we adapt the FTRL framework to control changes in the *trajectory occupancy measure*, rather than the conventional state-action occupancy measure used in earlier works. This is essential in our setting, as the occupancy induced by a policy must account for both the policy's history dependence and the adversarially changing transition dynamics.

Assuming access to accurate trajectory occupancies $q_{\pi, P_t}$ and trajectory loss function vector $\ell_t$, the ideal FTRL update rule with regularizer $\Phi(q_{\pi, P_t})$ is given by:

$$\pi_t = \arg\min_\pi f(\pi) := \sum_{\iota < t} \langle q_{\pi, P_\iota}, \ell_\iota \rangle + \frac{1}{\eta_t} \Phi(\bar{q}_{\pi, t}),$$

where $\bar{q}_{\pi, t} = \sum_{\iota < t} q_{\pi, P_\iota}$ represents the cumulative trajectory occupancy up to time $t$. However, in the *bandit feedback* setting with *unknown adversarial transitions*, we do not have access to the true occupancy or loss functions. Instead, we must rely on their estimators: $\sum_{\iota < t} \hat{q}_{\pi, P_\iota}$ and $\hat{\ell}_t$. A naive application of FTRL with these estimates inside the regularizer would lead to high variance and unstable updates. To mitigate this, we exploit the structure of the trajectory distribution and carefully rewrite the occupancy and regularization terms. Specifically, we decompose the averaged trajectory occupancy measure as:

$$\bar{q}_{\pi, t}(\tau) := \frac{1}{t-1} \sum_{\iota < t} q_{\pi, P_\iota} = \frac{1}{t-1} \Pi_\pi(\tau) \sum_{\iota < t} F_\iota(\tau) = \Pi_\pi(\tau) \bar{F}_t(\tau),$$

where we define $\Pi_\pi(\tau) = \prod_{k=0}^{K-1} \pi(a^k \mid h^k)$, $F_\iota(\tau) = P(s^0) \prod_{k=0}^{K-1} P_\iota(s^{k+1} \mid s^k, a^k)$ and $\{a^k\} \cup \{h^k\} \subset \tau$. This decomposition allows us to isolate the dependence on the policy $\pi$, which appears only in $\Pi_\pi(\tau)$, while treating $\bar{F}_t(\tau)$ as fixed with respect to policy optimization. This structure motivates our design of the regularization term $\Phi_t(\bar{q}_{\pi, t})$.

Regularization plays a central role in FTRL-style algorithms, directly influencing the convergence and stability of learning. However, since the distributional term $\bar{F}_t$ is unknown in our setting, we

define the regularizer using only the policy-dependent part. Specifically, we set:

$$\Phi_t(\bar{q}_{\pi,t}) = \sum_{\tau \in \mathcal{C}_\tau} \Pi_\pi(\tau) \log\left(\Pi_\pi(\tau)\right),$$

where $\Pi_\pi$ is the trajectory distribution induced by the policy $\pi$. This Shannon entropy regularizer encourages stability between successive policies while retaining theoretical tractability. Given this regularizer, we rewrite the FTRL update rule as:

$$\pi_t = \arg\min_\pi \sum_{\iota < t} \langle q_{\pi,P_\iota}, \ell_\iota \rangle + \frac{1}{\eta_t} \Phi_t(\bar{q}_{\pi,t}) = \arg\min_\pi \sum_{\iota < t} \sum_{\tau \in \mathcal{C}_\tau} \Pi_\pi(\tau) F_\iota(\tau) \ell_\iota(\tau) + \frac{1}{\eta_t} \Phi_t(\Pi_\pi)$$

$$= \arg\min_\pi \sum_{\iota < t} \langle \Pi_\pi, \mathcal{L}_\iota \rangle + \frac{1}{\eta_t} \Phi_t(\Pi_\pi) = \arg\min_\pi \langle \Pi_\pi, \Upsilon_t \rangle + \frac{1}{\eta_t} \Phi_t(\Pi_\pi),$$

where we define

$$\mathcal{L}_\iota(\tau) = F_\iota(\tau)\ell_\iota(\tau) = P(s_\iota^0) \prod_{k=0}^{K-1} P_\iota(s_\iota^{k+1} \mid s_\iota^k, a_\iota^k)\ell_\iota(\tau) \tag{1}$$

and $\Upsilon_t(\tau) = \sum_{\iota < t} \mathcal{L}_\iota(\tau)$.

This formulation allows us to perform efficient updates over policies while avoiding variance inflation from estimated transitions. Importantly, it also enables regret analysis under adversarial dynamics, as shown in the subsequent sections.

---

**Algorithm 1** Adversarial Dynamics Follow-the-Regularized-Leader (AD-FTRL) algorithm

---

1: **Initialize:** $\pi_0, \Upsilon_0, \Pi_{\pi_0}$
2: **for** $t = 0, 1, \ldots, T-1$ **do**
3:     Observe $s_t^0 \sim P_t^0(\cdot)$; Set $h_t^0 = \{s_t^0\}$
4:     **for** $k = 0, \ldots, K-1$ **do**
5:         Take the action: $a_t^k \sim \pi_t(\cdot \mid h_t^k)$
6:         Observe the loss $\ell_t(s_t^k, a_t^k)$, and next state $s_{t+1}^k \sim P_t(\cdot \mid s_t^k, a_t^k)$
7:     **end for**
8:     Observe the loss $\ell_t(s_t^K)$
9:     Get $\tau_t = \{s_t^0, a_t^0, \ldots, s_t^{K-1}, a_t^{K-1}, s_t^K\}$, $\ell_{\tau_t} = \sum_{h=0}^K \ell_t(s_t^k, a_t^k)$
10:     Update: $\widehat{\mathcal{L}}_t = \frac{\ell_{\tau_t}}{\gamma + \prod_{k=0}^{K-1} \pi_t(a_t^k \mid h_t^k)} [\mathbb{I}_{\tau=\tau_t}]_{\tau \in \mathcal{C}_\tau}$
11:     $\widehat{\Upsilon}_{t+1} = \widehat{\Upsilon}_t + \widehat{\mathcal{L}}_t$
12:     $\pi_{t+1} = \arg\min_{\pi \in \mathcal{C}_{\pi,\epsilon}} \left( \langle \Pi_\pi, \widehat{\Upsilon}_{t+1} \rangle + \frac{1}{\eta_{t+1}} \Phi_{t+1}(\Pi_\pi) \right)$
13: **end for**

---

Building on the discussions, we derived the adversarial dynamic FTRL update rule. Building on this foundation, we now present our AD-FTRL algorithm. The algorithm begins by initializing a history-dependent policy $\pi_0$, an estimated summation estimated trajectory-loss vector $\Upsilon_0$, and the corresponding vector $\Pi_{\pi_0}$.

At each episode $t$, the learner follows the policy $\pi_t$ to select actions, receives a loss, and observes a sampled trajectory. Since the policy $\pi_t$ is known, the associated distribution vector $\Pi_{\pi_t}$ can be computed directly. To update the summation trajectory-loss estimate, we use an unbiased estimator for $\widehat{\mathcal{L}}_t$, given by

$$\widehat{\mathcal{L}}_t = \frac{\ell_{\tau_t}}{\gamma + \prod_{k=0}^{K-1} \pi_t(a_t^k \mid h_t^k)} \cdot [\mathbb{I}_{\tau=\tau_t}]_{\tau \in \mathcal{C}_\tau}, \tag{2}$$

where $\gamma > 0$ is a small constant added for numerical stability and importance weight clipping. Next, the algorithm updates the cumulative vector $\Upsilon_{t+1}$ and computes the new policy $\pi_{t+1}$ according to the adversarial dynamics FTRL rule. To ensure the importance sampling estimator is unbiased, the updated policy $\pi_{t+1}$ is required to belong to an $\epsilon$-greedy history-dependent policy class $\mathcal{C}_{\pi,\epsilon}$, where

$\min \pi(\cdot) \geq \epsilon$. A typical choice is $\epsilon = \frac{1}{T}$. The regularizer used in the update is the Kullback–Leibler divergence,

$$\Phi_{t+1}(\Pi_\pi) = \sum_{\tau \in \mathcal{C}_\tau} \Pi_\pi(\tau) \log\left(\Pi_\pi(\tau)\right), \tag{3}$$

which encourages smooth updates in the trajectory distribution. This procedure is repeated iteratively for each episode.

**Remark 2.** *Compared with prior studies on FTRL-based algorithms, our approach shares the core principle of using FTRL to encourage smooth updates in the trajectory distribution. However, the key novelty in our method lies in applying the* trajectory occupancy measure—*rather than the traditional state-action occupancy measure—to constrain changes in the policy. This shift is crucial in the adversarial setting, where transition dynamics may vary arbitrarily. Moreover, we carefully design the regularization term to avoid dependence on unknown quantities, thereby preventing inflated variance in the regularizer and ensuring stable learning.*

# 5   THEORETICAL RESULTS: OPTIMAL MINIMAX REGRET BOUND

In this section, we first provide a sub-linear upper bound for our AD-FTRL algorithm. Next, we prove that our regret is indeed minimax optimal, i.e., matches the minimax lower bound we will show later.

**Theorem 5.1.** *For AD-FTRL algorithm, under adversarial bandit feedback loss functions and unknown adversarial transitions, set $\eta = \left(\frac{|\mathcal{S}|}{|\mathcal{A}|}\right)^{\frac{K}{2}} \log(|\mathcal{S}||\mathcal{A}|)\frac{1}{\sqrt{T}}, \gamma = \frac{1}{2(|\mathcal{S}||\mathcal{A}|)^{K/2}\sqrt{T}}$, it holds that*

$$Reg_T \leq \widetilde{\mathcal{O}}\left((|\mathcal{S}||\mathcal{A}|)^{K/2}\sqrt{T}\right).$$

Under the fully adversarial transitions setting, the corruption measure of transitions achieves the $\mathcal{O}(T)$ order. Under this setting, the regret bound in previous works (Jin et al., 2023) is linear in $T$ (where the regret bound is larger than $\mathcal{O}(C^P) = \mathcal{O}(T)$). But in our AD-FTRL algorithm, when the total episodes $T$ is larger enough, the sub-linear regret can be achieved.

Next, we provide the lower bound of this problem under fully unknown adversarial transitions and bandit feedback loss functions.

**Theorem 5.2** (Minimax Regret Lower Bound). *For any algorithm Alg, there exists an MDPs $\mathcal{M}_{Alg}^T = (\mathcal{S}, \mathcal{A}, K, \{P_t\}_{t\in[T]}, \{\ell_t\}_{t\in[T]})$, such that for $T > 4(|\mathcal{S}||\mathcal{A}|)^K \log T$, it holds that $Reg_T(Alg, \mathcal{M}_{Alg}^T) \geq \frac{\sqrt{(|\mathcal{S}||\mathcal{A}|)^K T}}{128} = \Omega\left(\sqrt{(|\mathcal{S}||\mathcal{A}|)^K T}\right)$; For $T \leq 4(|\mathcal{S}||\mathcal{A}|)^K \log T$, there exists $Reg_T(Alg, \mathcal{M}_{Alg}^T) \geq \widetilde{\Omega}(T)$.*

When $T \leq 4(|\mathcal{S}||\mathcal{A}|)^K \log T$, the regret upper bound is trivially $\mathcal{O}(T)$ due to bounded rewards. When $T > 4(|\mathcal{S}||\mathcal{A}|)^K \log T$, the regret lower bound matches the upper bound rate. Consequently, the rate $\min\{T, \sqrt{(|\mathcal{S}||\mathcal{A}|)^K T}\}$ characterizes the minimax-optimal regret for this adversarial RL setting. This result establishes that, without structural constraints on the transition dynamics, a regret of order $\min\{T, \sqrt{(|\mathcal{S}||\mathcal{A}|)^K T}\}$ is both unavoidable and achievable.

Under a related setting of Markov Games, (Tian et al., 2021) established a lower bound of $\sqrt{2^K T}$. However, their result does not specify dependence on the state or action space dimensions. In contrast, our result explicitly characterizes them in the complexity of learning under adversarial transition dynamics.

**Remark 3.** *In prior adversarial RL works, importance sampling is often avoided due to its high variance and its negative impact on regret bounds. However, our results demonstrate that, despite these challenges, importance sampling can still achieve the minimax-optimal regret rate in adversarial dynamic RL settings. We acknowledge that our FTRL-based algorithm may have limited applicability in practice, particularly in low-data regimes; however, it is still promising. First, the structure of history-dependent policies generalizes across settings, broadening the potential applications of our method. Moreover, with the great advancement of quantum computation and other computational resources, our algorithm has the potential to be applied more widely in practical reinforcement learning scenarios. For more details, the computational cost is provided in the appendix.*

## 6 PROOF SKETCH OF MINIMAX LOWER BOUND

In this section, we present the construction of a **hard MDPs instance** and provide a sketch of the minimax regret lower bound proof. We begin by describing the structure of the *hard MDPs*.

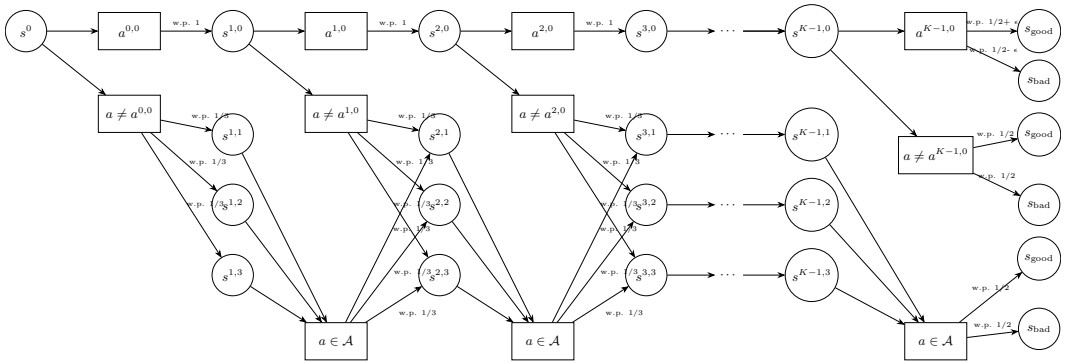

Figure 2: One of Hard MDPs: MDP with transitions $\mathbb{P}_\tau$, here the trajectory $\tau = \{s^0, a^{0,0}, s^{1,0}, a^{1,0}, ..., s^{K-1,0}, a^{K-1,0}, s^K = \text{good}\}$. $\ell(\text{good}) = 0, \ell(\text{bad}) = 1$, and $\ell(s^k, a^k) = 1$ if $k \neq K$ . The state transitions in this trajectory occur with probability 1. Other branches model deviations and are included with stochastic transitions (e.g., uniform distribution over next states when actions differ from those in the trajectory).

Let $\pi$ be a history-dependent policy. We define the set of successful trajectories under $\pi$ as $\Gamma(\pi) = \{\tau : s^K = \text{good} \in \tau, \Pi_\pi(\tau) \neq 0\}$. From Figure 2, we observe that each trajectory $\tau$ induces a corresponding MDP with transition kernel $\mathbb{P}_\tau$. To construct the minimax regret lower bound, we design a sequence of adversarial transition kernels. Given a deterministic history-dependent policy $\pi'$, we define, for each trajectory $\tau \in \Gamma(\pi')$, a corresponding transition kernel $\mathbb{P}_\tau$ using the process illustrated in Figure 2. Then, for each episode $t \in [T]$, the transition kernel $P_t$ is sampled uniformly from the set $\{\mathbb{P}_\tau : \tau \in \Gamma(\pi')\}$. The loss function is fixed and follows the structure defined in Figure 2. As a result, the MDP instance faced by the algorithm is drawn from the class:

$$\mathcal{M}_{\text{Alg}} \in \left\{ (\mathcal{S}, \mathcal{A}, K, \{P_t\}_{t \in [T]}, \ell) : P_t \in \{\mathbb{P}_\tau : \tau \in \Gamma(\pi^*)\} \right\}.$$

In this construction, the hard MDPs are designed such that the policy $\pi'$ is optimal for every MDP $(\mathcal{S}, \mathcal{A}, K, P_t, \ell)$ in the sequence. Based on this hard instance, an optimal history-dependent policy must select optimal actions using the observed trajectory history $h^{K-1}$. For histories $h^{K-1}$ that are prefixes of some $\tau \in \Gamma(\pi^*)$, choosing the correct action is critical. This reduces the problem to a contextual multi-armed bandit with $|\mathcal{S}|^K$ contexts and $|\mathcal{A}|$ arms—each corresponding to a possible history and action in the last stage—yielding a regret lower bound of order $\Omega(\sqrt{|\mathcal{S}|^K|\mathcal{A}|T})$.

To match the regret upper bound of our algorithm, however, we aim to prove a tighter lower bound of $\Omega(\sqrt{(|\mathcal{S}||\mathcal{A}|)^K T})$. Achieving this is non-trivial due to dependencies across different trajectories. For instance, if the algorithm knows that a trajectory belongs to $\Gamma(\pi')$, it can correctly infer parts of the optimal policy, such as $\pi'(\cdot \mid s^0)$, which introduces dependency between options. To formally establish the tighter lower bound, we reduce the problem to a composite hypothesis testing task. By leveraging Assouad's Lemma and Fano's method, we analyze the mutual information between the learner's observations and the composite hypothesis class (detailed proof is provided in the appendix).

## 7 CONCLUSION

In this paper, we first analyze the structural properties of the optimal policy under adversarially changing transitions and prove that the optimal policy must be *history-dependent*, instead of Markovian. Motivated by this observation, we introduce the concept of the *trajectory occupancy measure* and develop the **AD-FTRL** algorithm, which effectively operates under bandit feedback and adversarial transitions. We further show that our algorithm achieves a sublinear regret bound of order $\mathcal{O}\left(\sqrt{(|\mathcal{S}||\mathcal{A}|)^K T}\right)$, even in the fully adversarial setting, standing for the first sublinear result under

this setting. Furthermore, we construct an example and establish a matching lower bound, proving our regret is *minimax optimal*. These results collectively demonstrate both the necessity of handling history dependence and the fundamental difficulty of learning under adversarial transition dynamics. Our study provides the first comprehensive understanding of RL with adversarially changing transitions.

## ACAKNOWLEDGEMENT

The work of Y. Wang and S. Zou was supported in part by DARPA under Agreement No.D25AP00191, and by National Science Foundation under Grants ECCS-2438392(CAREER) and CCF-2438429. The work of K. Ji was generously supported by the NSF Career Award under Grant No. 2442418, NSF grants CCF-2311274 and ECCS-2326592.

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

## THE USE OF LARGE LANGUAGE MODELS (LLMS)

We used large language models (LLMs) to assist with phrasing, copy-editing, and LaTeX formatting. All technical choices, derivations, and results are our own; LLM outputs were reviewed and edited by the authors.

## A UPPER BOUND ANALYSIS.

### A.1 NOTATION AND PRELIMINARIES.

Firstly, we introduce essential notations used in our analysis.

We begin by reviewing the following definitions for a trajectory $\tau = \{s^0, a^0, \dots, s^K, a^K, s^{K+1}\}$ and histories $h^k = \{s^0, a^0, \dots, s^k\}$:

- $\Pi_t(\tau) := \Pi_{\pi_t}(\tau) = \prod_{k=0}^{K-1} \pi_t(a^k \mid h^k)$: the measure of generating trajectory $\tau$ under history-dependent policy $\pi_t$;
- $F_t(\tau) := F_{P_t}(\tau) = \prod_{k=0}^{K-1} P_t(s^{k+1} \mid s^k, a^k)$: the measure of transitioning along trajectory $\tau$ under transition kernel $P_t$;
- $\ell_t(\tau) = \sum_{k=0}^K \ell_t(s^k, a^k)$: the cumulative loss incurred along trajectory $\tau$;
- $\mathcal{L}_t(\tau) = F_t(\tau) \cdot \ell_t(\tau)$: the transition-weighted loss for trajectory $\tau$.

We also define the normalized trajectory loss estimator:

$$\widetilde{\mathcal{L}}_t(\tau) := \frac{\ell_t(\tau)}{\Pi_t(\tau)}.$$

This quantity is used in importance sampling when estimating the trajectory loss without requiring direct knowledge of the transition probabilities.

We now present a key lemma used in our analysis:

**Lemma A.1** (Trajectory Loss Estimation via Importance Sampling). *Let $\tau_t$ be a trajectory sampled under policy $\pi_t$ and transition kernel $P_t$, i.e., $\tau_t \sim q_{\pi_t, P_t}$. Then, for any $\tau \in \mathcal{C}_\tau$,*

$$\mathbb{E}_{\tau_t \sim q_{\pi_t, P_t}} \left[ \widetilde{\mathcal{L}}_t(\tau_t) \cdot \mathbb{I}_{\tau_t = \tau} \right] = \mathcal{L}_t(\tau).$$

This lemma shows that $\widetilde{\mathcal{L}}_t(\tau)$ is an unbiased estimator of $\mathcal{L}_t(\tau)$ when trajectories are sampled from $q_{\pi_t, P_t}$. Next, we define the cumulative trajectory loss:

$$\Upsilon_t(\tau) := \sum_{\iota \leq t} \mathcal{L}_\iota(\tau).$$

We use bold vector notation to represent trajectory-indexed quantities, such as: $\boldsymbol{\ell}_t = \{\ell_t(\tau)\}_{\tau \in \mathcal{C}_\tau}$, and likewise for $\Pi_t, \mathcal{L}_t, \Upsilon_t$, etc. With these notations in place, we now proceed to present the regret bound analysis for the proposed algorithm.

### A.2 PROOF OF THEOREM 5.1.

At the beginning, we introduce the high-probability upper bound of our AD-FTRL algorithm.

**Theorem A.2.** *For AD-FTRL algorithm, under adversarial bandit feedback loss functions and unknown adversarial transitions, set $\eta = \left(\frac{|\mathcal{S}|}{|\mathcal{A}|}\right)^{\frac{K}{2}} \log(|\mathcal{S}||\mathcal{A}|) \frac{1}{\sqrt{T}}, \gamma = \frac{1}{2(|\mathcal{S}||\mathcal{A}|)^{K/2}\sqrt{T}}$, with probability at least $1 - \delta$, it holds that*

$$Reg_T(\pi^*) = \sum_{t=1}^T V_t(\pi_t) - V_t(\pi^*)$$

$$\leq \frac{\log(1/\delta)}{2\gamma} + \gamma KT(|\mathcal{S}||\mathcal{A}|)^K + \eta K^2 T|\mathcal{A}|^K + K\eta \cdot \frac{\log(1/\delta)}{2\gamma} + \frac{1}{\eta}|\mathcal{S}|^K K \log(|\mathcal{S}||\mathcal{A}|T)$$

$$= \widetilde{\mathcal{O}}\left( \log(1/\delta)(|\mathcal{S}||\mathcal{A}|)^{\frac{K}{2}} \sqrt{T} \right).$$

Then, we will prove the Theorem 5.1. Firstly, we review the definitions that

$$V_t(\pi) := V_{P_t}(\pi) = \langle \Pi_\pi, \mathcal{L}_t \rangle.$$

Next, denote by estimated value function $\widetilde{V}_t = \langle \Pi_\pi, \widetilde{\mathcal{L}}_t \rangle$. Then, according to Lemma A.1,

$$\mathbb{E}[\widetilde{V}_t(\pi)] = \mathbb{E}\left[ \langle \Pi_\pi, \widetilde{\mathcal{L}}_t \rangle \right] = \langle \Pi_\pi, \mathcal{L}_t \rangle = V_t(\pi),$$

which means the $\widetilde{V}_t(\pi)$ is unbiased estimator of function $V_t(\pi)$ for any policy $\pi$.

Next, to simplex, we set $\Pi_* := \Pi_{\pi^*}, \Pi_t := \Pi_{\pi_t}$. We do the error decomposition of regret as follows:

$$\text{Reg}_T = \mathbb{E}\left[ \sum_{t=1}^{T} V_t(\pi_t) - V_t(\pi^*) \right] = \mathbb{E}\left[ \sum_{t=1}^{T} \langle \Pi_{\pi_t} - \Pi_*, \mathcal{L}_t \rangle \right]$$

$$= \underbrace{\mathbb{E}\left[ \sum_{t=1}^{T} \langle \Pi_{\pi_t} - \Pi_*, \widehat{\mathcal{L}}_t \rangle \right]}_{\text{error reg}} + \underbrace{\mathbb{E}\left[ \sum_{t=1}^{T} \langle \Pi_{\pi_t}, \mathcal{L}_t - \widehat{\mathcal{L}}_t \rangle \right]}_{\text{error 1}} + \underbrace{\mathbb{E}\left[ \sum_{t=1}^{T} \langle \Pi_*, \widehat{\mathcal{L}}_t - \mathcal{L}_t \rangle \right]}_{\text{error 2}}. \quad (4)$$

**Error 1 Term:**

Firstly, it holds that error $1 < 0$. The $\widetilde{\mathcal{L}} =$ defined is unbiased estimator, and $\mathbb{E}[\sum_{t=1}^{T} \langle \Pi_{\pi_t}, \mathcal{L}_t - \widetilde{\mathcal{L}}_t \rangle] = 0$. Since it holds that

$$\widehat{\mathcal{L}}_t = \frac{\ell_{\tau_t}}{\gamma + \prod_{k=0}^{K-1} \pi_t(a_t^k \mid s_t^k)} \cdot [\mathbb{I}_{\tau=\tau_t}]_{\tau \in \mathcal{C}_\tau}$$

$$= \frac{\ell_{\tau_t} \prod_{k=0}^{K-1} \pi_t(a_t^k \mid s_t^k) \cdot [\mathbb{I}_{\tau=\tau_t}]_{\tau \in \mathcal{C}_\tau}}{(\gamma + \prod_{k=0}^{K-1} \pi_t(a_t^k \mid s_t^k)) \prod_{k=0}^{K-1} \pi_t(a_t^k \mid s_t^k)}$$

$$\geq -\frac{\gamma \ell_{\tau_t} \cdot [\mathbb{I}_{\tau=\tau_t}]_{\tau \in \mathcal{C}_\tau}}{(\gamma + \prod_{k=0}^{K-1} \pi_t(a_t^k \mid s_t^k)) \prod_{k=0}^{K-1} \pi_t(a_t^k \mid s_t^k)} + \frac{\ell_{\tau_t} \cdot [\mathbb{I}_{\tau=\tau_t}]_{\tau \in \mathcal{C}_\tau}}{\prod_{k=0}^{K-1} \pi_t(a_t^k \mid s_t^k)}$$

$$= -\frac{\gamma \ell_{\tau_t} \cdot [\mathbb{I}_{\tau=\tau_t}]_{\tau \in \mathcal{C}_\tau}}{(\gamma + \Pi_{\pi_t}(\tau))\Pi_{\pi_t}(\tau)} + \widetilde{\mathcal{L}}_t,$$

where inequality follows from that

$$(\gamma + \prod_{k=0}^{K-1} \pi_t(a_t^k \mid s_t^k))\ell_{\tau_t} [\mathbb{I}_{\tau=\tau_t}]_{\tau \in \mathcal{C}_\tau} - \gamma \ell_{\tau_t} [\mathbb{I}_{\tau=\tau_t}]_{\tau \in \mathcal{C}_\tau}$$

$$= \prod_{k=0}^{K-1} \pi_t(a_t^k \mid s_t^k) \cdot \ell_{\tau_t} [\mathbb{I}_{\tau=\tau_t}]_{\tau \in \mathcal{C}_\tau}.$$

Next, we can bound the error 1 as:

$$\mathbb{E}\left[ \sum_{t=1}^{T} \langle \Pi_{\pi_t}, \mathcal{L}_t - \widehat{\mathcal{L}}_t \rangle \right] \leq \mathbb{E}\left[ \sum_{t=1}^{T} \langle \Pi_{\pi_t}, \mathcal{L}_t - \widetilde{\mathcal{L}}_t \rangle \right] - \mathbb{E}\left[ \sum_{t=1}^{T} \frac{-\gamma \ell_{\tau_t} \cdot \mathbb{I}_{\tau=\tau_t}}{(\gamma + \Pi_{\pi_t}(\tau))} \right]$$

$$= \sum_{t=1}^{T} \sum_{\tau} \frac{\gamma \ell_{\tau_t} \cdot \Pi_{\pi_t}(\tau)}{\gamma + \Pi_{\pi_t}(\tau)}$$

$$\leq \gamma \sum_{t=1}^{T} \sum_{\tau} K = \gamma K T (|\mathcal{S}||\mathcal{A}|)^K, \quad (5)$$

where inequality follows from the fact that $\ell_{\tau_t} \leq K$.

**Error 2 Term:**

Then, for term error 2, we introduce the following lemma:

**Lemma A.3.** *For the sequence of vectors $\alpha_1, \ldots, \alpha_T$, $\alpha_t \in [0, 2\gamma]^{(|\mathcal{S}||\mathcal{A}|)^H}$, we have with probability at least $1 - \delta$,*

$$\sum_{t=1}^{T} \sum_{\tau \in \mathcal{C}_\tau} \alpha_t(\tau) \left( \widehat{\mathcal{L}}_t(\tau) - \mathcal{L}_t(\tau) \right) \leq \log \left( \frac{1}{\delta} \right).$$

From the Lemma A.1 and the fact $\Pi(\tau) \leq 1$ and $2\gamma\Pi(\tau) \leq 2\gamma$ for all $\Pi, \tau$, we can easily get that with probability at least $1 - \delta$,

$$\text{error } 2 = \mathbb{E} \left[ \sum_{t=1}^{T} \langle \Pi_*, \widehat{\mathcal{L}}_t - \mathcal{L}_t \rangle \right] = \frac{1}{2\gamma} \mathbb{E} \left[ \sum_{t=1}^{T} \langle 2\gamma\Pi_*, \widehat{\mathcal{L}}_t - \mathcal{L}_t \rangle \right] \leq \frac{\log(\frac{1}{\delta})}{2\gamma}; \tag{6}$$

**Error Reg Term:**

Next, we analyze the error reg term. Firstly, we introduce a lemma for the convexity of the trajectory distribution set.

**Lemma A.4.** *The set $\{\Pi_\pi : \pi \in \mathcal{C}_\pi\}$ forms a convex set.*

According to the AD-FTRL update rule,

$$\Pi_{t+1} = \arg\min_{\Pi_\pi, \pi \in \mathcal{C}_\pi} \eta_t \langle \Pi_\pi, \sum_{\iota \leq t} \widehat{\mathcal{L}}_\iota \rangle + \Phi_t(\Pi_\pi) = \arg\min_{\Pi_\pi, \pi \in \mathcal{C}_\pi} \eta_t \langle \Pi_\pi, \widehat{\Upsilon}_t \rangle + \Phi_t(\Pi_\pi).$$

Then, set $\eta_1 = \eta_2, \ldots, = \eta_T = \eta$. Denote the Bregman divergence with convex function $F$, i.e. $D_f(p, q) = F(p) - F(q) - \langle p - q, \nabla F(q) \rangle$. Combined with Lemma A.4, we can get the follows equation:

$$\langle \Pi_t, \sum_{\iota < t} \widehat{\mathcal{L}}_\iota \rangle + \frac{1}{\eta} \Phi_t(\Pi_t)$$

$$= \langle \Pi_{t+1}, \sum_{\iota < t} \widehat{\mathcal{L}}_\iota \rangle + \frac{1}{\eta} \Phi_t(\Pi_{t+1}) - \left( \langle \Pi_{t+1} - \Pi_t, \sum_{\iota < t+1} \widehat{\mathcal{L}}_\iota \rangle - \frac{1}{\eta} \Phi_t(\Pi_t) + \frac{1}{\eta} \Phi_t(\Pi_{t+1}) \right)$$

$$\leq \langle \Pi_{t+1}, \sum_{\iota < t} \widehat{\mathcal{L}}_\iota \rangle + \frac{1}{\eta} \Phi_t(\Pi_{t+1}) - \left( \left\langle \Pi_{t+1} - \Pi_t, \frac{1}{\eta} \nabla \Phi_t(\Pi_t) \right\rangle - \frac{1}{\eta} \Phi_t(\Pi_t) + \frac{1}{\eta} \Phi_t(\Pi_{t+1}) \right)$$

$$= \langle \Pi_{t+1}, \sum_{\iota < t} \widehat{\mathcal{L}}_\iota \rangle + \frac{1}{\eta} \Phi_t(\Pi_{t+1}) - D_{\frac{1}{\eta}\Phi_t}(\Pi_{t+1}, \Pi_t)$$

$$= \langle \Pi_{t+1}, \sum_{\iota < t+1} \widehat{\mathcal{L}}_\iota \rangle + \frac{1}{\eta} \Phi_{t+1}(\Pi_{t+1}) - \langle \Pi_{t+1}, \widetilde{\mathcal{L}}_t \rangle + \frac{1}{\eta} \Phi_t(\Pi_{t+1}) - \frac{1}{\eta} \Phi_{t+1}(\Pi_{t+1}) - D_{\frac{1}{\eta}\Phi_t}(\Pi_{t+1}, \Pi_t),$$

where the inequality comes from and convex property in Lemma A.4 and the first order optimality condition of $\Pi_t$, i.e. $\left\langle \Pi_{t+1} - \Pi_t, \sum_{\iota < t} \widetilde{\mathcal{L}}_\iota + \frac{1}{\eta} \nabla \Phi_t(\Pi_t) \right\rangle \geq 0$.

Taking the summation of above equation and we can get that

$$\langle \Pi_1, \sum_{\iota < 1} \widehat{\mathcal{L}}_\iota \rangle + \frac{1}{\eta} \Phi_1(\Pi_1) \leq \langle \Pi_T, \sum_{\iota < T+1} \widehat{\mathcal{L}}_\iota \rangle + \frac{1}{\eta} \Phi_{T+1}(\Pi_{T+1})$$

$$- \sum_{t=1}^{T} \langle \Pi_{t+1}, \widetilde{\mathcal{L}}_t \rangle - \sum_{t=1}^{T} \left( \frac{1}{\eta} \Phi_{t+1}(\Pi_{t+1}) - \frac{1}{\eta} \Phi_t(\Pi_{t+1}) \right) - \sum_{t=1}^{T} D_{\frac{1}{\eta}\Phi_t}(\Pi_{t+1}, \Pi_t).$$

Here, $\langle \Pi_1, \sum_{\iota < 1} \widehat{\mathcal{L}}_\iota \rangle = 0$. Then, we can rewrite the error reg term as follows:

$$\sum_{t=1}^{T} \langle \Pi_t - \Pi_*, \widehat{\mathcal{L}}_t \rangle \leq \sum_{t=1}^{T} \langle \Pi_t - \Pi_*, \widehat{\mathcal{L}}_t \rangle + \left\langle \Pi_{T+1}, \sum_{\iota=1}^{T} \widehat{\mathcal{L}}_\iota \right\rangle + \frac{1}{\eta} \Phi_{T+1}(\Pi_{T+1}) - \frac{1}{\eta} \Phi_1(\Pi_1)$$

$$- \sum_{t=1}^{T} \langle \Pi_{t+1}, \widetilde{\mathcal{L}}_t \rangle - \sum_{t=1}^{T} \left( \frac{1}{\eta} \Phi_{t+1}(\Pi_{t+1}) - \frac{1}{\eta} \Phi_t(\Pi_{t+1}) \right) - \sum_{t=1}^{T} D_{\frac{1}{\eta}\Phi_t}(\Pi_{t+1}, \Pi_t)$$

$$\leq \sum_{t=1}^{T} \left( \langle \Pi_t - \Pi_{t+1}, \widehat{\mathcal{L}}_t \rangle - D_{\frac{1}{\eta}\Phi_t}(\Pi_{t+1}, \Pi_t) \right) - \sum_{t=1}^{T} \langle \Pi_*, \widehat{\mathcal{L}}_t \rangle + \left\langle \Pi_{T+1}, \sum_{t=1}^{T} \widehat{\mathcal{L}}_t \right\rangle + \frac{1}{\eta} \Phi_{T+1}(\Pi_{T+1})$$

$$- \frac{1}{\eta} \Phi_1(\Pi_1) - \frac{1}{\eta} \sum_{t=1}^{T} (\Phi_{t+1}(\Pi_{t+1}) - \Phi_t(\Pi_{t+1}))$$

$$\leq \underbrace{\sum_{t=1}^{T} \left( \langle \Pi_t - \Pi_{t+1}, \widehat{\mathcal{L}}_t \rangle - D_{\frac{1}{\eta}\Phi_t}(\Pi_{t+1}, \Pi_t) \right) + \frac{1}{\eta} \left( \Phi_{T+1}(\Pi_*) - \Phi_1(\Pi_1) \right)}_{\text{Term I}}$$

$$\underbrace{- \frac{1}{\eta} \sum_{t=1}^{T} (\Phi_{t+1}(\Pi_{t+1}) - \Phi_t(\Pi_{t+1}))}_{\text{Term II}}, \tag{7}$$

where the last inequality follows from the fact that:

$$\left\langle \Pi_{T+1}, \sum_{t=1}^{T} \widehat{\mathcal{L}}_t \right\rangle + \frac{1}{\eta} \Phi_{T+1}(\Pi_{T+1}) \leq \left\langle \Pi_*, \sum_{t=1}^{T} \widehat{\mathcal{L}}_t \right\rangle + \frac{1}{\eta} \Phi_{T+1}(\Pi_*).$$

**Boundary of Term I:**

Next, we bound Term I. We relax the constraint $\Pi \in \mathcal{R}^{|\mathcal{S}||\mathcal{A}|}$ and get the following inequality:

$$\langle \Pi_t - \Pi_{t+1}, \widehat{\mathcal{L}}_t \rangle - D_{\frac{1}{\eta}\Phi_t}(\Pi_{t+1}, \Pi_t) \leq \max_{\Pi \in \mathcal{R}^{(|\mathcal{S}||\mathcal{A}|)^K}} \left\{ \langle \Pi_t - \Pi, \widehat{\mathcal{L}}_t \rangle - D_{\frac{1}{\eta}\Phi_t}(\Pi, \Pi_t) \right\}$$

Next, we define the optimal value vector $\widetilde{\Pi}_t$ as follows:

$$\widetilde{\Pi}_t := \arg\max_{\Pi} \left\{ \eta \langle \Pi_t - \Pi, \widehat{\mathcal{L}}_t \rangle - D_{\frac{1}{\eta}\Phi_t}(\Pi, \Pi_t) \right\}.$$

Here, we need to remark that the condition $\widetilde{\Pi}_{t+1} \notin \{\Pi_\pi : \pi \in \mathcal{C}_\pi\}$ may happen.

We begin by analyzing the optimal condition of the update, i.e.

$$\frac{\partial}{\partial \Pi} \left( \eta \langle \Pi_t - \Pi, \widehat{\mathcal{L}}_t \rangle - D_{\Phi_t}(\Pi \| \Pi_t) \right) = 0.$$

We compute the derivative of the Bregman divergence:

$$\frac{\partial}{\partial \Pi} D_{\Phi_t}(\Pi \| \Pi_t) = \frac{\partial}{\partial \Pi} \left( \Phi_t(\Pi) - \Phi_t(\Pi_t) - \langle \Pi - \Pi_t, \nabla\Phi_t(\Pi_t) \rangle \right) = \nabla\Phi_t(\Pi) - \nabla\Phi_t(\Pi_t).$$

Therefore, it holds that

$$\nabla\Phi_t(\widetilde{\Pi}_t) - \nabla\Phi_t(\Pi_t) = -\eta\widehat{\mathcal{L}}_t.$$

Using the identity $\nabla\Phi_t(\Pi) = \langle \Pi, \log \frac{\Pi}{\Pi_t} \rangle = \log \frac{\Pi}{\Pi_t} + 1$,, we can obtain that:

$$\nabla\Phi_t(\widetilde{\Pi}_t) - \nabla\Phi_t(\Pi_t) = \log \frac{\widetilde{\Pi}_t}{\Pi_t} = -\eta\widehat{\mathcal{L}}_t.$$

Above, we can get the expression of $\widetilde{\Pi}_t$, i.e.,

$$\widetilde{\Pi}_t = \Pi_t \cdot \exp(-\eta\widehat{\mathcal{L}}_t).$$

Next, substituting the definition of $\Phi_t$, we analyze the Bregman divergence.

$$
\begin{aligned}
D_{\Phi_t}(\Pi \,\|\, \Pi_t) &= \Phi_t(\Pi) - \Phi_t(\Pi_t) - \langle \Pi - \Pi_t, \nabla \Phi_t(\Pi_t) \rangle \\
&= \langle \Pi, \log \Pi \rangle - \langle \Pi_t, \log \Pi_t \rangle - \langle \Pi - \Pi_t, \log \Pi_t + 1 \rangle \\
&= \left\langle \Pi, \log \frac{\Pi}{\Pi_t} \right\rangle - \Pi + \Pi_t.
\end{aligned}
\tag{8}
$$

We now continue the regret analysis by bounding the inner product term. Using the optimality of $\widetilde{\Pi}_t$, we upper bound the above expression by:

$$
\begin{aligned}
\eta \langle \Pi_t - \Pi_{t+1}, \widehat{\mathcal{L}}_t \rangle - D_{\Phi_t}(\Pi_{t+1}, \Pi_t) &\leq \eta \langle \Pi_t - \widetilde{\Pi}_t, \widehat{\mathcal{L}}_t \rangle - D_{\Phi_t}(\widetilde{\Pi}_t, \Pi_t) \\
&= \langle \Pi_t - \widetilde{\Pi}_t, \eta \widehat{\mathcal{L}}_t \rangle - \Phi_t(\widetilde{\Pi}_t) + \Phi_t(\Pi_t) - \langle \widetilde{\Pi}_t - \Pi_t, \nabla \Phi_t(\Pi_t) \rangle \\
&\overset{(a)}{=} \langle \Pi_t - \widetilde{\Pi}_t, -\nabla \Phi_t(\widetilde{\Pi}_t) \rangle - \Phi_t(\widetilde{\Pi}_t) + \Phi_t(\Pi_t) \\
&= D_{\Phi_t}(\Pi_t, \widetilde{\Pi}_t),
\end{aligned}
\tag{9}
$$

where step (a) uses the fact that

$$
\nabla \Phi_t(\Pi_t) - \nabla \Phi_t(\widetilde{\Pi}_t) = \eta \widehat{\mathcal{L}}_t; \widetilde{\Pi} = \Pi_t \exp(-\eta \widehat{\mathcal{L}}_t).
$$

We now bound the term I from Eq. 7. Recall that:

$$
\Pi_t(\tau) \cdot \widehat{\mathcal{L}}_t(\tau) = \Pi_t(\tau) \cdot \frac{\ell_t(\tau) \cdot \mathbb{I}_{\tau=\tau}}{\gamma + \Pi_t(\tau)} \leq \max \ell_t(\tau) \leq K.
$$

Using this, the cumulative Bregman divergence is bounded as:

$$
\begin{aligned}
\sum_{t=1}^{T} D_{\Phi_t}(\Pi_t, \widetilde{\Pi}_t) &= \left\langle \Pi_t, \log \frac{\widetilde{\Pi}_t}{\Pi_t} \right\rangle - \widetilde{\Pi}_t + \Pi_t \\
&= \Pi_t \left( \eta \widehat{\mathcal{L}}_t + \exp(-\eta \widehat{\mathcal{L}}_t) - 1 \right) \\
&\leq \sum_{t=1}^{T} \sum_{\tau \in \mathcal{C}_\tau} \Pi_t(\tau) \cdot \eta^2 \cdot \widehat{\mathcal{L}}_t(\tau)^2 \leq K \sum_{t=1}^{T} \sum_{\tau \in \mathcal{C}_\tau} \eta^2 \cdot \widehat{\mathcal{L}}_t(\tau),
\end{aligned}
$$

where the first inequality comes from the fact that $y + e^{-y} - 1 \leq y^2$ for all $y > -1$ and $\eta \widehat{\mathcal{L}}_t \geq -1$.

We now invoke a concentration result. Let $\alpha_t(\tau) = 2\gamma$, combined with Lemma A.3, with probability at least $1 - \delta$, we have:

$$
\sum_{t=1}^{T} \sum_{\tau \in \mathcal{C}_\tau} 2\gamma \cdot (\widehat{\mathcal{L}}_t(\tau) - \mathcal{L}_t(\tau)) \leq \log \left( \frac{1}{\delta} \right).
\tag{10}
$$

With probability at least $1 - \delta$, combining these gives

$$
\begin{aligned}
\sum_{t=1}^{T} D_{\Phi_t}(\Pi_t, \widetilde{\Pi}_t) &\leq K \sum_{t=1}^{T} \sum_{\tau \in \mathcal{C}_\tau} \eta^2 \cdot \widehat{\mathcal{L}}_t(\tau) \\
&= K \sum_{t=1}^{T} \sum_{\tau \in \mathcal{C}_\tau} \eta^2 \cdot \mathcal{L}_t(\tau) + K \eta^2 \sum_{t=1}^{T} \sum_{\tau \in \mathcal{C}_\tau} (\widehat{\mathcal{L}}_t(\tau) - \mathcal{L}_t(\tau)) \\
&\overset{(a)}{\leq} K \sum_{t=1}^{T} \sum_{\tau \in \mathcal{C}_\tau} \eta^2 \cdot \mathcal{L}_t(\tau) + K \eta^2 \cdot \frac{\log(1/\delta)}{2\gamma} \\
&\overset{(b)}{=} K \sum_{t=1}^{T} \sum_{\tau \in \mathcal{C}_\tau} \eta^2 \cdot \left( \prod_{k=0}^{K-1} P_t(s^{k+1} \mid s^k, a^k) \right) \cdot \ell_t(\tau) + K \eta^2 \cdot \frac{\log(1/\delta)}{2\gamma} \\
&\leq K^2 T \eta^2 \cdot |\mathcal{A}|^K + K \eta^2 \cdot \frac{\log(1/\delta)}{2\gamma},
\end{aligned}
$$

where $(a)$ follows from the equation 10 and $(b)$ follows from the definition of $\mathcal{L}$ in equation 1.

According to the above equation and equation 9, with probability at least $1 - \delta$, we obtain the regret bound:

$$\sum_{t=1}^{T} \left( \langle \Pi_t - \Pi_{t+1}, \widehat{\mathcal{L}}_t \rangle - D_{\frac{1}{\eta} \Phi_t}(\Pi_{t+1}, \Pi_t) \right) \leq K^2 T \eta \cdot |\mathcal{A}|^K + K\eta \cdot \frac{\log(1/\delta)}{2\gamma}.$$

**Boundary of Term I:**

We consider the Term II: $\sum_{t=1}^{T} \left( \Phi_t(\Pi_{t+1}) - \Phi_{t+1}(\Pi_{t+1}) \right)$ in Eq. 7. From the definition of regularizer term in equation 3, it holds that

$$\Phi_{t+1}(\Pi_{t+1}) - \Phi_t(\Pi_{t+1}) = 0.$$

Therefore, Term II $= 0$.

**Boundary of the Regularizer Terms:**

We next bound $\Phi_t(\Pi$ for any $\Pi$ and $t$ by a constant, where $\Pi \in \{\Pi_\pi : \pi \text{ is } \frac{1}{T}\text{-greedy policy class}\}$. Then, it holds that

$$\Phi_t(\Pi) = \sum_\tau \Pi(\tau) \log\left(\Pi(\tau)\right) \leq \sum_\tau \Pi(\tau) \log\left(\frac{1}{|\mathcal{S}||\mathcal{A}|T}\right)^K = K \log(|\mathcal{S}||\mathcal{A}|T) \cdot |\mathcal{S}|^K.$$

Therefore, with probability at least $1 - \delta$, the error reg term can be bounded as:

$$\sum_{t=1}^{T} \langle \Pi_t - \Pi_*, \widehat{\mathcal{L}}_t \rangle \leq \eta K^2 T |\mathcal{A}|^K + \eta K \cdot \frac{\log(1/\delta)}{2\gamma} + \frac{1}{\eta} |\mathcal{S}|^K K \log(|\mathcal{S}||\mathcal{A}|T). \tag{11}$$

Above all, combining all error 1 bound in equation 5, error 2 in equation 6, and error reg in equation 11, we obtain that with probability at least $1 - \delta$:

$$\sum_{t=1}^{T} V_t(\pi_t) - V_t(\pi^*) \leq \frac{\log(1/\delta)}{2\gamma} + \gamma K T (|\mathcal{S}||\mathcal{A}|)^K$$

$$+ \eta K^2 T |\mathcal{A}|^K + K\eta \cdot \frac{\log(1/\delta)}{2\gamma} + \frac{1}{\eta} |\mathcal{S}|^K K \log(|\mathcal{S}||\mathcal{A}|T) \tag{12}$$

**Bounding deviation in expected regret.** We also observe that the difference in expected regret from replacing $\pi^*$ with $\pi_t$ is bounded as:

$$\sum_{t=1}^{T} V_t(\pi_t) - V_t(\pi^*) \leq \sum_{t=1}^{T} V_t(\pi_t) \leq KT.$$

Now setting $\delta = \frac{1}{T}$, we bound the expected regret:

$$\text{Reg}_T = \mathbb{E}\left[\sum_{t=1}^{T} V_t(\pi_t) - V_t(\pi^*)\right]$$

$$\leq K + \frac{\log T}{2\gamma} + \gamma K T (|\mathcal{S}||\mathcal{A}|)^K + \eta T |\mathcal{A}|^K K^2 + K\eta \cdot \frac{\log T}{2\gamma} + \frac{1}{\eta} |\mathcal{S}|^K K \log(|\mathcal{S}||\mathcal{A}|T).$$

Optimizing the learning rate $\eta$ and sampling rate $\gamma$. Set the parameters as follows:

$$\eta = \left(\frac{|\mathcal{S}|}{|\mathcal{A}|}\right)^{K/2} \log(|\mathcal{S}||\mathcal{A}|) \cdot \frac{1}{\sqrt{T}}, \qquad \gamma = \frac{1}{2\left(|\mathcal{S}||\mathcal{A}|\right)^{K/2} \sqrt{T}}.$$

Then the regret becomes:

$$\text{Reg}_T \leq K + (|\mathcal{S}||\mathcal{A}|)^{K/2} \sqrt{T} \log T + (|\mathcal{S}||\mathcal{A}|)^{K/2} \sqrt{T} \frac{K}{2}$$

$$+ K^2 \left(|\mathcal{S}||\mathcal{A}|\right)^{K/2} \log(|\mathcal{S}||\mathcal{A}|) + K \left(|\mathcal{S}||\mathcal{A}|\right)^{K/2} \cdot \sqrt{T} \log T + (|\mathcal{S}||\mathcal{A}|)^{K/2} \cdot \sqrt{T} \log T$$

$$\leq \mathcal{O}\left((|\mathcal{S}||\mathcal{A}|)^{K/2} \sqrt{T}\right).$$

A.3 DISCUSSION: ALTERNATIVE REGULARIZATION VIA KL AND MIRROR DESCENT.

We briefly discuss the effect of switching the regularizer. In particular, one can replace our default regularization with a Kullback–Leibler (KL) term and perform mirror descent instead of AD-FTRL algorithm. Concretely, given the similar step, the KL-regularized mirror step is defined as:

$$\pi_{t+1} \;=\; \underset{\pi \in \mathcal{C}_\pi}{\arg\max} \left\{ \eta_t \langle g_t, \Pi_\pi \rangle - \Phi_{KL}(\Pi_\pi \| \Pi_t) \right\},$$

where regularization term $\Phi_{\mathrm{KL}}(\Pi_\pi \| \Pi_t)$ plays the role of a KL-type divergence,

$$\Phi_{t+1}(\Pi_\pi) := \Phi_{\mathrm{KL}}(\Pi_\pi \| \Pi_t) = \left\langle \Pi_\pi, \log\left( \frac{\Pi_\pi}{\Pi_t} \right) \right\rangle.$$

Although it resembles the KL divergence, it is only *KL-like* since $\Pi_\pi$ is an occupancy measure rather than a normalized distribution.

**Remark 4.** *This formulation can be viewed as mirror descent with the negative entropy of occupancy measures as the mirror map, where $\Phi_{\mathrm{KL}}$ serves as the corresponding Bregman divergence. Intuitively, it balances exploitation of the gradient direction $g_t$ with proximity to the previous iterate $\Pi_t$ in the KL geometry. From a theoretical standpoint, the analysis follows the same high-level structure as in our main framework (via one-step potential inequalities and telescoping arguments). However, the non-normalization of $\Pi_\pi$ introduces additional technical difficulties, making the analysis considerably more involved. These challenges are addressed in detail in the above proof.*

---

**Algorithm 2** Adversarial Dynamics Mirror Descent (AD-MD) algorithm

---

1: **Initialize:** $\pi_0, \Upsilon_0, \Pi_{\pi_0}$
2: **for** $t = 0, 1, \ldots, T-1$ **do**
3:     Observe $s_t^0 \sim P_t^0(\cdot)$; Set $h_t^0 = \{s_t^0\}$
4:     **for** $k = 0, \ldots, K-1$ **do**
5:         Take the action: $a_t^k \sim \pi_t(\cdot \mid h_t^k)$
6:         Observe the loss $\ell_t(s_t^k, a_t^k)$, and next state $s_{t+1}^k \sim P_t(\cdot \mid s_t^k, a_t^k)$
7:     **end for**
8:     Observe the loss $\ell_t(s_t^K)$
9:     Get $\tau_t = \{s_t^0, a_t^0, \ldots, s_t^{K-1}, a_t^{K-1}, s_t^K\}$, $\ell_{\tau_t} = \sum_{h=0}^K \ell_t(s_t^k, a_t^k)$
10:     Update: $\widehat{\mathcal{L}}_t = \frac{\ell_{\tau_t}}{\gamma + \prod_{k=0}^{K-1} \pi_t(a_t^k|h_t^k)} [\mathbb{I}_{\tau=\tau_t}]_{\tau \in \mathcal{C}_\tau}$
11:     $\pi_{t+1} = \arg\min_{\pi \in \mathcal{C}_{\pi,\epsilon}} \left( \langle \Pi_\pi, \widehat{\mathcal{L}}_t \rangle + \frac{1}{\eta_{t+1}} \Phi_{t+1}(\Pi_\pi) \right)$
12: **end for**

---

### A.3.1 COMPUTATIONAL COST DISCUSSION

Here we discuss the computational cost of the algorithms. The major computational cost lies in the Line 12 of Algorithm 1 and Line 11 of Algorithm 2. In Algorithm 1, the computational cost relies in solving an optimization problem defined over a space of size $(|\mathcal{S}||\mathcal{A}|)^K$. This step dominates the overall computational cost of the algorithm.

In the Algorithm 2, the Line 11 can be rewritten as the policy optimization (PO)-based method.

By applying the KKT conditions and noting that $\widehat{\mathcal{L}}_t$ contains only one nonzero entry, we obtain the following update:

$$\pi_{t+1}(a^k|h^k) \propto \pi_t(a^k|h^k) \exp(-\eta_t \widehat{\mathcal{L}}_t).$$

Hence, for all $a, h, k$, the policy can be iteratively updated as:

$$\pi_{t+1}(a^k|h^k) = \frac{\pi_t(a^k|h^k) \exp(-\eta_t \ell_{\tau_t} \mathbb{I}(\{h^k, a^k\} \in \tau_t))}{\sum_{a \in \mathcal{A}^k} \pi_t(a|h^k) \exp(-\eta_t \ell_{\tau_t} \mathbb{I}(\{h^k, a\} \in \tau_t))}.$$

Under this formulation, the computational cost is reduced from $\mathcal{O}((|\mathcal{S}||\mathcal{A}|)^K)$ to $\mathcal{O}(|\mathcal{S}||\mathcal{A}|K)$. Therefore, the proposed **PO-based method** offers a more efficient alternative.

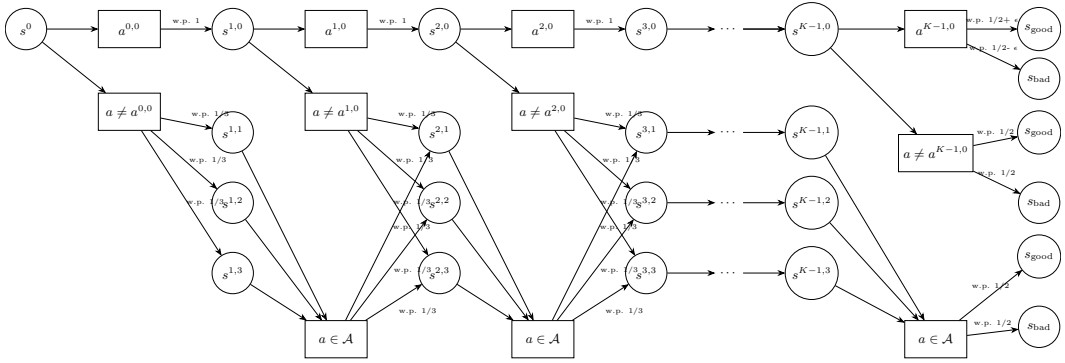

Figure 3: Illustration of the MDP $\mathbb{P}_\tau$, $\tau = \{s^0, a^{0,0}, s^{1,0}, a^{1,0}, ..., s^{K-1,0}, a^{K-1,0}\}$. The top row shows the trajectory induced by a deterministic history-dependent policy $\pi_\theta$. The state transitions in this trajectory occur with probability 1. Other branches model deviations and are included with stochastic transitions (e.g., uniform distribution over next states when actions differ from those in the trajectory). The loss is received only in the terminal states, i.e., $\ell(G) = 0$ and $\ell(B) = 1$, where $G$ denotes the "good" state and $B$ denotes the "bad" state.

# B LOWER BOUND ANALYSIS.

## B.1 HARD MDPs DESIGN.

In this section, we provide the detailed Hard MDPs design process for further proof.

Let $\pi$ be a history-dependent policy. We define the set of successful trajectories under $\pi$ as $\Gamma(\pi) = \{\tau : s^K = \text{good} \in \tau, \Pi_\pi(\tau) \neq 0\}$. From Figure 2, we observe that each trajectory $\tau$ induces a corresponding MDP with transition kernel $\mathbb{P}_\tau$, i.e.

$$\mathbb{P}_\tau(s^{k+1}|s^k, a^k) = 1, \text{ if } \{s^k, a^k, s^{k+1}\} \subset \tau;$$

$$\mathbb{P}_\tau(s^{k+1}|s^k, a^k) = \frac{1}{|\mathcal{S}| - 1}, \text{ if } s^k \in \tau, a^k \notin \tau, s^{k+1} \notin \tau;$$

$$\mathbb{P}_\tau(s^{k+1}|s^k, a^k) = \frac{1}{|\mathcal{S}|}, \text{ if } s^k \notin \tau.$$

To construct the minimax regret lower bound, we design a sequence of adversarial transition kernels. Given a deterministic history-dependent policy $\pi'$, we define, for each trajectory $\tau \in \Gamma(\pi')$, a corresponding transition kernel $\mathbb{P}_\tau$ using the process illustrated in Figure 2. Then, for each episode $t \in [T]$, the transition kernel $P_t$ is sampled uniformly from the set $\{\mathbb{P}_\tau : \tau \in \Gamma(\pi')\}$. The loss function is fixed and follows the structure defined in Figure 2. As a result, the MDP instance faced by the algorithm is drawn from the class:

$$\mathcal{M}_{\text{Alg}} \in \left\{ (\mathcal{S}, \mathcal{A}, K, \{P_t\}_{t \in [T]}, \ell) : P_t \in \{\mathbb{P}_\tau : \tau \in \Gamma(\pi^*)\} \right\}.$$

In this construction, the hard MDPs are designed such that the policy $\pi'$ is optimal for every MDP $(\mathcal{S}, \mathcal{A}, K, P_t, \ell)$ in the sequence. Based on this hard instance, an optimal history-dependent policy must select optimal actions using the observed trajectory history $h^{K-1}$. For histories $h^{K-1}$ that are prefixes of some $\tau \in \Gamma(\pi^*)$, choosing the correct action is critical. This reduces the problem to a multi-armed bandit with $|\mathcal{S}|^K$ arms—each corresponding to a possible history—yielding a regret lower bound of order $\Omega(\sqrt{|\mathcal{S}|^K T})$, which is lower than the target minimax optimal lower bound.

## B.2 MOTIVATION

In this section, we introduce the proof sketch and motivation. At the beginning, we introduce theorems concerning the regret and loss lower bound of parameter estimation.

**Theorem B.1** (Theorem 31.2 (Assouad's lemma) (Polyanskiy & Wu, 2025)). *Assume that the loss function $\ell$ satisfies the $\alpha$-triangle inequality*

$$\ell(\theta_0, \theta_1) \leq \alpha \left( \ell(\theta_0, \theta) + \ell(\theta_1, \theta) \right).$$

*Suppose $\Theta$ contains a subset $\Theta' = \{\theta_b : b \in \{0,1\}^d\}$ indexed by the hypercube, such that $\ell(\theta_b, \theta_{b'}) \geq \beta \cdot d_k(b, b')$ for all $b, b'$ and some $\beta > 0$. Then*

$$\inf_{\hat{\theta}} \sup_{\theta \in \Theta} \mathbb{E}_\theta \ell(\theta, \hat{\theta}) \geq \frac{\beta d}{4\alpha} \left( 1 - \max_{d_k(b,b')=1} \mathrm{TV}(P_{\theta_b}, P_{\theta_{b'}}) \right). \tag{13}$$

**Theorem B.2** (Theorem 31.3 (Polyanskiy & Wu, 2025)). *Let $d$ be a metric on $\Theta$. Fix an estimator $\hat{\theta}$. For any $T \subset \Theta$ and $\epsilon > 0$,*

$$\mathbb{P}\left[ d(\theta, \hat{\theta}) \geq \frac{\epsilon}{2} \right] \geq 1 - \frac{C(T) + \log 2}{\log M(T, d, \epsilon)}, \tag{14}$$

*where $C(T) \triangleq \sup I(\theta; X)$ is the capacity of the channel from $\theta$ to $X$ with input space $T$, with the supremum taken over all distributions (priors) on $T$. Consequently,*

$$\inf_{\hat{\theta}} \sup_{\theta \in \Theta} \mathbb{E}_\theta[d(\theta, \hat{\theta})^r] \geq \sup_{T \subset \Theta, \, \epsilon > 0} \left( \frac{\epsilon}{2} \right)^r \left( 1 - \frac{C(T) + \log 2}{\log M(T, d, \epsilon)} \right). \tag{15}$$

In our work, we aim to provide the lower bound with order $\mathcal{O}((|\mathcal{S}||\mathcal{A}|)^{\frac{K}{2}} \sqrt{T})$. For this tighter lower bound, simple Fona's Two points method is not enough. Therefore, we combine the above two theorems to provide the regret lower bound.

### B.2.1 DESIGN $\theta$ AND $\theta_b$

Firstly, we parameterize the history-dependent deterministic policy $\pi_\theta$, here $\theta \in \mathbb{R}^{(|\mathcal{S}||\mathcal{A}|)^K}$. We prove that there exists a one-to-one mapping from policy $\pi$ to parameter $\theta$, i.e. , for $\tau \in \mathcal{C}_\tau$, denote by $\theta[\tau]$ the $\tau$ entry of the vector $\theta$, then, the mapping from policy $\pi$ to parameter vector $\theta$ is:

$$\theta_\pi[\tau] = \prod_{k=0}^{K-1} \pi(a^k|h^k), \quad \pi_\theta(a^k|h^k) = \mathbb{1}\left( \sum_{\{a^k\}, h^k \subset \tau} \theta_\tau \neq 0 \right)$$

Based on the above definition, we define the parameter set $\Theta$ as follows:

$$\Theta := \{\theta : \pi_\theta \text{ is a deterministic history dependent policy}\}$$

Next, we introduce the *policy optimal trajectories set* as follows:

$$\Gamma(\theta) = \Gamma(\pi_\theta), \quad \Gamma(\pi) = \{\tau' \mid \Pi(\tau', \pi) \neq 0, \tau' \in \mathcal{C}\},$$

where

$$\Pi(\tau, \pi) = \prod_{k=0}^{K-1} \pi(a^k|h^k) \mathbb{1}[\{a^k\}, h^k \subset \tau].$$

*Remark:* If the trajectory $\tau \in \Gamma(\pi_\theta)$, then under our constructed hard MDPs $\mathcal{M}_\theta$, the expected loss of $\tau$ is minimized. This implies that any sampled trajectory belonging to $\Gamma(\pi_\theta)$ corresponds to an optimal trajectory.

Next, denote trs $= (s^0, s^1, \ldots, s^K) \in \mathcal{S}^K$ the *state trace*. Then, we can find a *one-to-one* mapping $[|\mathcal{S}|^K] \to \mathcal{C}_{\mathrm{trs}} = \bigotimes_{k=1}^{K} \mathcal{S}_k$, for any number $n \in [|\mathcal{S}|^K]$,

$$\mathcal{T}_{\mathrm{trs}}(n) = \mathrm{trs}, \text{ s.t. } \mathcal{T}^{-1}(\mathrm{trs}) = n,$$

where $\mathcal{C}_{\mathrm{trs}}$ is the state trace available set and function $\mathcal{T}^{-1}$ is the inverse function of $\mathcal{T}_{\mathrm{trs}}$.

Next, we introduce a function to find a corresponding trajectory, i.e.,

$$\Gamma(\mathcal{T}_{\mathrm{trs}}(d), \theta) \Rightarrow \tau, \quad \text{s.t. } \prod_{k=0}^{K-1} \pi_\theta(a^k \mid h^k) = 1, \quad \mathcal{T}_{\mathrm{trs}}(d) \subset \tau,$$

where a state trace trs $\subset \tau$ indicates that all states in trs appear in the trajectory $\tau$ in the same order.

One transition kernel $\mathbb{P}_\tau$ is constructed based on the transition kernel generated by the trajectory shown in the first row of Figure 3.

Specifically, the set $\Gamma(\pi_\theta)$ includes all $|\mathcal{S}|^K$ deterministic trajectories that are consistent with the policy $\pi_\theta$. Each such trajectory generates a distinct transition kernel.

Then, **hard MDPs** $\mathcal{M}_\theta$ are defined using these transition kernels, sampled uniformly from the set of all kernels induced by $\Gamma(\pi_\theta)$. Formally, the transition probability of $\mathcal{M}_\theta$ is given by:

$$\tau \sim \mathcal{M}_\theta \text{ from } \Gamma(\pi_\theta); \mathbb{P}_\tau(s^{k+1} \mid s^k, a^k) \text{ for } k = 0, 1, ..., K-1$$

where $P_\tau$ is the transition kernel induced by trajectory $\tau$, $\mathcal{M}_\theta$ is uniform distribution over $\Gamma(\pi_\theta)$. Here, the transition kernel in one episode is assumed to be the same.

**Remark 5.** *Here, we discuss the adversarial or disturbed nature of the MDP process: in each episode, the MDP samples a transition kernel from a reliability set. The transition kernel remains fixed throughout the episode but may change between episodes.*

**Remark 6.** *The hard MDP $\mathcal{M}_\theta$ has the following properties:*

- $\pi_\theta \in \arg\min_\pi V_{\mathbb{P}_\tau}(\pi)$, *here* $\mathbb{P}_\tau \in \mathcal{M}_\theta$. *Then, set* $\bar{V}_{\mathcal{M}_\theta}(\pi) = \mathbb{E}_{\mathbb{P}_\tau \sim \mathcal{M}_\theta}[V_{\mathbb{P}_\tau}(\pi)]$, $\pi_\theta \in \arg\min_\pi \bar{V}_{\mathcal{M}_\theta}(\pi)$.

- $V_{\mathbb{P}_\tau}(\pi_\theta) = \frac{1}{2} - \epsilon$ *and* $\bar{V}_{\mathcal{M}_\theta}(\pi_\theta) = \frac{1}{2} - \epsilon$.

Therefore, under the **hard MDPs** $\mathcal{M}_\theta$, optimal history dependent policy is $\pi_\theta$. The sample complexity analysis problem can be transferred to the following estimation process $\theta \to \mathcal{M}_\theta \to I^T \to \hat{\theta}$. Here, $I^t$ is the histories up to episode $t$ the $\mathcal{I}^t = \{I^t\}$ is the set of possible histories up to episode $t$. The algorithm tries to estimate the value of $\theta$ from the sampled histories $I^T$. However, the regret-bound analysis is more challenging. The problem should be transferred to the following estimation iteration: $\theta \to (\mathcal{M}_\theta, \hat{\theta}_t) \to I^t \to \hat{\theta}_{t+1}$. In general, the sample complexity lower bound analysis is fundamental to the regret lower bound.

Therefore, let us review the Theorems B.1 and B.2. From these two theorems, we could see that it is the first step to design a mapping from $b$ to $\theta_b$ in our work, where $b \in \{0, 1\}^{|\mathcal{S}|^K}$. Since $b$ is a $0-1$ vector, a general idea for designing a mapping is to set $\bar{\theta}$ as a baseline.

Then, we define a function to find the corresponding trajectory:

$$\Gamma^c(\mathcal{T}_{\text{trs}}(d), \theta) \Rightarrow \tau^c, \quad \text{s.t.} \quad \left(\prod_{k=0}^{k-2} \pi_\theta(a^k \mid h^k)\right) \cdot \pi_\theta(A_{-1}^{k-1}(a_1^k) \mid h^{k-1}) = 1,$$

where $h^k \cup \{a^{k-1}\} \subset \tau^c$. Here, set the function $A^k(\cdot)$ as follows:

$$A^k = \{a^{k,1}, \dots, a^{k,|\mathcal{A}^k|}\}, \quad A^k(a^{k,i}) = \begin{cases} a^{k,i-1}, & i \neq 1 \\ a^{k,|\mathcal{A}^k|}, & i = 1 \end{cases}$$

Next consider any $b \in \{0, 1\}^{|\mathcal{S}|^K}, \bar{\theta} \in \Theta, d \in [|\mathcal{S}|^K]$

$$g(d, b_d = 0, \theta) := \Gamma(\mathcal{T}_{\text{trs}}(d), \theta), \quad g(d, b_d = 1, \theta) := \Gamma^c(\mathcal{T}_{\text{trs}}(d), \theta)$$

Set a $\pi_{\theta'}$ as follows:
$$\pi_{\theta'}(a^k \mid h^k) = \pi_\theta(a^k \mid h^k), \quad \text{for } k < K-1$$

$$\pi_{\theta'}(a_1^{k-1} \mid h^{k-1}) = \begin{cases} 1, & \text{if } a_1^{k-1} \in g(d, b_d, \theta) \\ 0, & \text{otherwise} \end{cases} \quad \text{where } d = \mathcal{N}(\text{trs}(h^{k-1}))$$

Then, it holds that
$$\theta_{\bar{\theta}, b} = \theta(\pi_{\theta'})$$

Define the parameters set: $\Theta_{\bar{\theta}} := \left\{\theta_{b, \bar{\theta}} \mid b \in \{0, 1\}^{|\mathcal{S}|^K}\right\}$; $\mu_b$: uniform distribution over $\{0, 1\}^{|\mathcal{S}|^K}$.

Above all, we design a set $\mathcal{M}_{\bar{\theta}} = (\Theta_{\bar{\theta}}, \mu_b)$. By revising Theorem B.1, the problem can be reduced to estimating $\hat{\theta}$ from the set $\Theta_{\bar{\theta}}$. Applying Theorem B.1 and B.2, we could obtain sample complexity lower bound. However, establishing a regret lower bound requires further analysis, as discussed earlier.

## B.3 Episode-History Analysis

Now, define the episode information $I^t = i^t = \{\tau_t, \ell_t(\tau_t)\}$, where $\ell_t$ is the total loss at step $t$. From the Figure 3, the loss structure is defined as:

$$\ell(G) = 0, \qquad \ell(B) = 1.$$

That is, the loss distribution only depends on the trajectory $\tau_t$, i.e.,

$$\mathbb{P}(\ell_t = \cdot \mid \tau_t) = \mathbb{P}(\ell_t = \cdot \mid i^{t-1}, \tau_t),$$

meaning the loss at step $t$ depends only on $\tau_t$.

Here, we define $\nu_t^k = i^{t-1} \cup \{s_t^0, a_t^0, \ldots, s_t^k\}$, and use $\mathrm{Alg}_t^k(\cdot \mid \nu_t^k) \in \Delta(\mathcal{A})$ to represent the algorithm's action at step $k$ of round $t$. Here, $\mathrm{Alg}_t^k(\cdot \mid \nu_t^k)$ denotes the distribution over actions induced by the algorithm. Then, the joint probability of observing a sequence of episode information $i^T$ under the transition kernel sequence (which is unknown for the estimator) $P^1, \ldots, P^T$ is:

$$\mathbb{P}(I^T = i^T \mid P_1, \ldots, P_T) = \prod_{t=1}^{T} \left( \prod_{k=0}^{K-1} \mathrm{Alg}_t^k(a_t^k \mid \nu_t^k) \cdot P_t^k(s_t^{k+1} \mid s_t^k, a_t^k) \right) P_t(\ell_t \mid \tau_t)$$

$$= \prod_{t=1}^{T} \left( \prod_{k=0}^{K-1} \mathrm{Alg}_t^k(a_t^k \mid \nu_t^k) \right) \cdot \tilde{P}_t(\tau_t) P_t(\ell_t \mid \tau_t),$$

where $\nu_t^k = i^{t-1} \cup \{s_t^0, a_t^0, \ldots, s_t^k\}$ and $\tilde{P}_t(\tau_t) = \prod_{k=0}^{K-1} P_t^k(s_t^{k+1} \mid s_t^k, a_t^k)$.

Now, take expectation over the randomness of the transition kernel sequence under prior $\mathcal{M}_\theta$:

$$\mathbb{P}_{\mathcal{M}_\theta}(I^T = i^T) = \mathbb{E}_{P_1, \ldots, P_T \sim \mathcal{M}_\theta} \left[ \mathbb{P}(I^T = i^T \mid P_1, \ldots, P_T) \right]$$

$$= \mathbb{E}_{P_1, \ldots, P_T \sim \mathcal{M}_\theta} \left[ \prod_{t=1}^{T} \left( \prod_{k=0}^{K-1} \mathrm{Alg}_t^k(a_t^k \mid \nu_t^k) \right) \tilde{P}_t(\tau_t) P_t(\ell_t \mid \tau_t) \right]$$

$$= \prod_{t=1}^{T} \left( \prod_{k=0}^{K-1} \mathrm{Alg}_t^k(a_t^k \mid \nu_t^k) \right) \cdot \underbrace{\mathbb{E}_{P_1, \ldots, P_T \sim \mathcal{M}_\theta} \left[ \prod_{t=1}^{T} \tilde{P}_t(\tau_t) P_t(\ell_t \mid \tau_t) \right]}_{\text{term } I}.$$

Now consider the inner expectation term:

$$\text{term } I = \mathbb{E}_{P_1, \ldots, P_T \sim \mathcal{M}_\theta} \left[ \mathbb{E}_{P_1, \ldots, P_T \sim \mathcal{M}_\theta} \left[ \prod_{t=1}^{T} \tilde{P}_t(\tau_t) P_t(\ell_t \mid \tau_t) \mid P_1, \ldots, P_T \right] \right]$$

$$= \mathbb{E}_{P_1, \ldots, P_{T-1} \sim \mathcal{M}_\theta} \left[ \prod_{t=1}^{T-1} \tilde{P}_t(\tau_t) P_t(\ell_t \mid \tau_t) \cdot \mathbb{E}_{P_T \sim \mathcal{M}_\theta}[\tilde{P}_T(\tau_T) P_T(r_T \mid \tau_T)] \right]. \qquad (16)$$

**Lemma B.3.** *Define the joint distribution*

$$\tilde{P}_{\mathcal{M}_\theta}(\tau) := \mathbb{E}_{P \sim \mathcal{M}_\theta} \left[ \tilde{P}(\tau) \right] \text{ and } \tilde{P}_{\mathcal{M}_\theta}(\tau, \ell) := \mathbb{E}_{P \sim \mathcal{M}_\theta} \left[ \tilde{P}(\tau) P(\ell \mid \tau) \right],$$

*where* $\tilde{P}(\tau) = \prod_{k=0}^{K-1} P(s^{k+1} \mid s^k, a^k)$, $s^k, a^k, s^{k+1} \in \tau$.

*Then, it holds that*

$$\tilde{P}_{\mathcal{M}_\theta}(\tau) = \frac{1}{|\mathcal{S}|^K};$$

$$\tilde{P}_{\mathcal{M}_\theta}(\tau, \ell = 1) = \begin{cases} \frac{1}{|\mathcal{S}|^K} \left( \frac{1}{2} - \epsilon \right), & \text{if } \tau \in \Gamma(\pi_\theta) \\ \frac{1}{|\mathcal{S}|^K} \cdot \frac{1}{2}, & \text{otherwise.} \end{cases}$$

Recall the expression for term I in Eq. 16:

$$\text{Term I} = \mathbb{E}_{P_1,\dots,P_T \sim \mathcal{M}_\theta} \left[ \prod_{t=1}^{T} \tilde{P}_t(\tau_t) P_t(\ell_t \mid \tau_t) \right]$$

$$= \mathbb{E}_{P_1,\dots,P_T \sim \mathcal{M}_\theta} \left[ \prod_{t=1}^{T} \tilde{P}_t(\tau_t) P_t(\ell_t \mid \tau_t) \right] \tilde{P}_{\mathcal{M}_\theta}(\tau_T, \ell_T)$$

$$= \dots = \prod_{t=1}^{T} \tilde{P}_{\mathcal{M}_\theta}(\tau_t, \ell_t).$$

We then obtain the full marginal probability:

$$\mathbb{P}_{\mathcal{M}_\theta}(I^T = i^T) = \prod_{t=1}^{T} \left( \prod_{k=0}^{K-1} \text{Alg}_t^k(a_t^k \mid \nu_t^k) \right) \cdot \tilde{P}_{\mathcal{M}_\theta}(\tau_t, \ell_t),$$

where

$$\tilde{P}_{\mathcal{M}_\theta}(\tau_t, \ell_t = 0) = \begin{cases} \frac{1}{|\mathcal{S}|^K} \left( \frac{1}{2} + \epsilon \right), & \tau_t \in \Gamma(\pi_\theta), \\ \frac{1}{|\mathcal{S}|^K} \cdot \frac{1}{2}, & \text{otherwise.} \end{cases}$$

### B.4   REGRET ANALYSIS

Firstly, given a fixed $I^T = i^T$, the regret is defined as:

$$\text{Reg}\left(I^T = i^T, \mathcal{M}_\theta\right) = \sum_{t=1}^{T} \ell_t - \sum_{t=1}^{T} V_t(\pi^*)$$

$$= \sum_{\tau \in \Gamma(\pi_\theta)} \sum_{t=1}^{T} \mathbb{I}(\tau_t = \tau)\mathbb{I}(\ell_t = 0) - (\tfrac{1}{2} - \epsilon)T.$$

Now consider the expected regret:

$$\text{Reg}(\mathcal{M}_\theta) = \mathbb{E}\left[\text{Reg}\left(I^T = i^T, \mathcal{M}_\theta\right)\right]$$

$$= \mathbb{E}\left[ \sum_{\tau \in \Gamma(\pi_\theta)} \sum_{t=1}^{T} \mathbb{I}(\tau_t = \tau)\mathbb{I}(\ell_t = 0) \right] - (\tfrac{1}{2} - \epsilon)T$$

$$= \sum_{\tau \in \Gamma(\pi_\theta)} \sum_{t=1}^{T} \mathbb{E}_{I^T \sim \mathbb{P}_{\mathcal{M}_\theta}} \left[ \mathbb{I}(\tau_t = \tau)\mathbb{I}(\ell_t = 0) \right] - (\tfrac{1}{2} - \epsilon)T -$$

$$\overset{(a)}{=} \sum_{\tau \in \Gamma(\pi_\theta)} \mathbb{E}_{I^T \sim \mathbb{P}_{\mathcal{M}_\theta}} \left[ \sum_{t=1}^{T} \mathbb{I}(\tau_t = \tau)\mathbb{P}_{\mathcal{M}_\theta}(\ell_t = 1 \mid \tau_t = \tau) \right] - (\tfrac{1}{2} - \epsilon)T$$

$$= \sum_{\tau \in \Gamma(\pi_\theta)} \mathbb{E}_{I^T \sim \mathbb{P}_{\mathcal{M}_\theta}} \left[ \sum_{t=1}^{T} \mathbb{I}(\tau_t = \tau) \right] \cdot (\tfrac{1}{2} - \epsilon) - (\tfrac{1}{2} - \epsilon)T$$

$$- \sum_{\tau \notin \Gamma(\pi_\theta)} \mathbb{E}_{I^T \sim \mathbb{P}_{\mathcal{M}_\theta}} \left[ \sum_{t=1}^{T} \mathbb{I}(\tau_t = \tau) \right] \cdot \tfrac{1}{2}$$

$$= \epsilon T - \epsilon \sum_{\tau \in \Gamma(\pi_\theta)} \mathbb{E}_{I^T \sim \mathbb{P}_{\mathcal{M}_\theta}} \left[ \sum_{t=1}^{T} \mathbb{I}(\tau_t = \tau) \right],$$

where $(a)$ follows from the fact that if $\tau_t \in \Gamma(\pi_\theta)$, then

$$\mathbb{P}(\ell_t = 1 \mid \tau_t) = \mathbb{P}(\ell_t = 0 \mid i^{t-1}, \tau_t) = \tfrac{1}{2} - \epsilon,$$

otherwise,

$$\mathbb{P}(\ell_t = 1 \mid \tau_t) = \mathbb{P}(\ell_t = 0 \mid i^{t-1}, \tau_t) = \tfrac{1}{2}.$$

Next, define $\mathcal{N}_\tau = \mathcal{N}_\tau(I^T) := \sum_{t=1}^T \mathbb{I}(\tau_t = \tau)$. Then, it holds that:

$$\mathrm{Reg}_{\mathcal{M}_\theta} \leq \epsilon \left[ T - \sum_{\tau \in \Gamma(\pi_\theta)} \mathbb{E}_{I^T \sim \mathbb{P}_{\mathcal{M}_\theta}} \left[ \mathcal{N}_\tau(I^T) \right] \right].$$

## B.5 REGRET LOWER BOUND PROOF: $\mathcal{O}(|\mathcal{S}|^{\frac{K}{2}} \sqrt{T})$.

Consider a fixed $\bar{\theta}, b \in \{0, 1\}^{|\mathcal{S}|^K}$, and define

$$\Theta_{\bar{\theta}} := \{\theta_b, \bar{\theta} : b \in \{0, 1\}^{|\mathcal{S}|^K}\},$$

where $\mu_b(b = b) = \frac{1}{2^{|\mathcal{S}|^K}}$.

Then, it holds that

$$\max_{\theta \in \Theta} \mathrm{Reg}(\mathcal{M}_\theta) \geq \mathbb{E}_{\mu_b} \left[ \mathrm{Reg}\big(\mathcal{M}(\theta_{b,\bar{\theta}})\big) \right]$$

$$= \mathbb{E}_{\mu_b} \left[ \epsilon T - \epsilon \sum_{\tau \in \Gamma(\pi_{\theta_{b,\bar{\theta}}})} \mathbb{E}_{I^T \sim \mathbb{P}_{\mathcal{M}(\theta_{b,\bar{\theta}})}} \left[ \mathcal{N}_\tau(I^T) \right] \right]$$

$$= \epsilon T - \epsilon \mathbb{E}_{\mu_b} \left[ \sum_{\tau \in \Gamma(\pi_{\theta_{b,\bar{\theta}}})} \mathbb{E}_{I^T \sim \mathbb{P}_{\mathcal{M}(\theta_{b,\bar{\theta}})}} \left[ \mathcal{N}_\tau(I^T) \right] \right].$$

Next, let us review the definition. The function to find the corresponding trajectory is defined as:

$$\Gamma(\mathcal{T}_{\mathrm{trs}}(d), \theta) \Rightarrow \tau, \quad \text{s.t.} \quad \prod_{k=0}^{K-1} \pi_\theta(a^k \mid h^k) = 1, \quad \mathcal{T}_{\mathrm{trs}}(d) \subset \tau,$$

$$\Gamma^c(\mathcal{T}_{\mathrm{trs}}(d), \theta) \Rightarrow \tau^c, \quad \text{s.t.} \quad \left( \prod_{k=0}^{k-2} \pi_\theta(a^k \mid h^k) \right) \cdot \pi_\theta(A_{-1}^{k-1}(a_1^k) \mid h^{k-1}) = 1,$$

where $h^k \cup \{a^{k-1}\} \subset \tau^c$. Here, the function $A^k(\cdot)$ is defined as follows:

$$A^k = \{a^{k,1}, \ldots, a^{k,|\mathcal{A}^k|}\}, \quad A^k(a^{k,i}) = \begin{cases} a^{k,i-1}, & i \neq 1 \\ a^{k,|\mathcal{A}^k|}, & i = 1 \end{cases}$$

Next consider any $b \in \{0, 1\}^{|\mathcal{S}|^K}, \bar{\theta} \in \Theta, d \in [|\mathcal{S}|^K]$

$$g(d, b_d = 0, \theta) := \Gamma(\mathcal{T}_{\mathrm{trs}}(d), \theta), \quad g(d, b_d = 1, \theta) := \Gamma^c(\mathcal{T}_{\mathrm{trs}}(d), \theta).$$

Then, we could get that:

$$\sum_{\tau \in \Gamma(\pi_{\bar{\theta}})} \mathbb{E}_{I^T} \left[ \mathcal{N}_\tau(I^T) \right] = \sum_{d=1}^{|\mathcal{S}|^K} \mathbb{E}_{I^T} \left[ \mathcal{N}_{g(d, b_d, \bar{\theta})} \right]. \tag{17}$$

For any $\theta_b$, $g(d, b_d, \bar{\theta}) \in \Gamma(\pi_{\theta_{b,\bar{\theta}}})$. If $\tau = g(d, b_d, \bar{\theta})$, then $\prod_{k=0}^{K-1} \pi_{\bar{\theta}}(a^k \mid h^k) = 1$ and $h^k \subset \tau$.

Since $|\{\tau = g(d, b_d, \bar{\theta}) : d \in [|\mathcal{S}|^k]\}| = |\mathcal{S}|^k$, and all such $\tau \subset I$, the set is well-defined.

Apply Eq.17 to the last inequality, it holds that

$$\max_{\theta} \text{Reg}(\mathcal{M}_{\theta}) \geq \epsilon T - \epsilon \mathbb{E}_{\mu_b} \left[ \sum_{\tau \in \Gamma(\pi_{\theta_{b,\bar{\theta}}})} \mathbb{E}_{I^T \sim \mathbb{P}_{\mathcal{M}(\theta_{b,\bar{\theta}})}} \left[ \mathcal{N}_{\tau}(I^T) \right] \right]$$

$$= \epsilon \left( T - \mathbb{E}_{\mu_b} \left[ \sum_{d=1}^{|\mathcal{S}|^K} \mathbb{E}_{I^T} [\mathcal{N}_{g(d,b_d,\bar{\theta})}] \right] \right)$$

$$= \epsilon \left( T - \frac{1}{2} \sum_{d=1}^{|\mathcal{S}|^K} \left( \mathbb{E}_{\mu_b | b_d = 1} [\mathcal{N}_{g(d,b_d=1,\bar{\theta})}] + \mathbb{E}_{\mu_b | b_d = 0} [\mathcal{N}_{g(d,b_d=0,\bar{\theta})}] \right) \right),$$

where we used that $\mu_b(b_d = 0) = \mu_b(b_d = 1) = \frac{1}{2}$.

Define the joint distribution of histories $I^T$ under conditional marginal:

$$\mathbb{P}(I^T \mid b_d = 1) := \mathbb{E}_{\mu_b} \left[ \mathbb{P}(I^T \mid \mathcal{M}_{\theta_{b,\bar{\theta}}}, b_d = 1, \bar{\theta}) \right].$$

Then, it holds that

$$\max_{\theta} \text{Reg}(\mathcal{M}_{\theta}) \geq \epsilon T - \frac{\epsilon}{2} \sum_{d=1}^{|\mathcal{S}|^K} \left( \mathbb{E}_{\mathbb{P}(I^T | b_d = 1, \bar{\theta})} [\mathcal{N}_{g(d,b_d=1,\bar{\theta})}] + \mathbb{E}_{\mathbb{P}(I^T | b_d=0,\bar{\theta})} [\mathcal{N}_{g(d,b_d=0,\bar{\theta})}] \right).$$

Next, define

$$\mathcal{N}_{g(d,\bar{\theta})} = \mathcal{N}_{g(d,b_d=1,\bar{\theta})} + \mathcal{N}_{g(d,b_d=0,\bar{\theta})}.$$

Set event:

$$A_d := \left\{ \mathcal{N}_{g(d,\bar{\theta})} \leq \frac{2T}{|\mathcal{S}|^k} \right\}.$$

**Lemma B.4** (Bounding Visit Counts and Concentration). *If $T > 2|\mathcal{S}|^{\frac{K}{2}} \sqrt{T \log T}$, then $1 - \mathbb{P}(A_d) \leq \frac{1}{T^3}$ for any algorithm.*

Next, consider the following upper bound term that arises in the regret lower bound derivation.

$$\frac{1}{2} \sum_{d=1}^{|\mathcal{S}|^K} \mathbb{E}_{\mathbb{P}(I^T | b_d=1)} [\mathcal{N}_{g(d,b_d=1,\bar{\theta})}] + \mathbb{E}_{\mathbb{P}(I^T | b_d=0)} [\mathcal{N}_{g(d,b_d=0,\bar{\theta})}].$$

This term quantifies the expected number of visits to the constructed trajectories under both possible values of the binary indicator $b_d$. We decompose this using the indicator event $A_d := \{ \mathcal{N}_{g(d,\bar{\theta})} \leq \frac{2T}{|\mathcal{S}|^K} \}$:

$$\frac{1}{2} \sum_{d=1}^{|\mathcal{S}|^K} \mathbb{E}_{\mathbb{P}(I^T | b_d=1)} [\mathcal{N}_{g(d,b_d=1,\bar{\theta})}] + \mathbb{E}_{\mathbb{P}(I^T | b_d=0)} [\mathcal{N}_{g(d,b_d=0,\bar{\theta})}]$$

$$= \frac{1}{2} \sum_{d=1}^{|\mathcal{S}|^K} \left[ \mathbb{E}_{\mathbb{P}(I^T | b_d=1)} [\mathcal{N}_{g(d,b_d=1,\bar{\theta})} \mathbb{I}(A_d)] + \mathbb{E}_{\mathbb{P}(I^T | b_d=0)} [\mathcal{N}_{g(d,b_d=0,\bar{\theta})} \mathbb{I}(A_d)] \right]$$

$$+ \frac{1}{2} \sum_{d=1}^{|\mathcal{S}|^K} \left[ \mathbb{E}_{\mathbb{P}(I^T | b_d=1)} [\mathcal{N}_{g(d,b_d=1,\bar{\theta})} \mathbb{I}(A_d^c)] + \mathbb{E}_{\mathbb{P}(I^T | b_d=0)} [\mathcal{N}_{g(d,b_d=0,\bar{\theta})} \mathbb{I}(A_d^c)] \right].$$

We now bound the second term. Since the number of visits $\mathcal{N}_g \leq T$ deterministically and the event complement probability $\mathbb{P}(A_d^c) \leq \frac{1}{T^3}$ by Lemma B.4, we get:

$$\frac{1}{2} \sum_{d=1}^{|\mathcal{S}|^K} \left[ \mathbb{E}_{\mathbb{P}(I^T | b_d = 1)}[\mathcal{N}_{g(d, b_d = 1, \bar{\theta})} \mathbb{I}(A_d^c)] + \mathbb{E}_{\mathbb{P}(I^T | b_d = 0)}[\mathcal{N}_{g(d, b_d = 0, \bar{\theta})} \mathbb{I}(A_d^c)] \right]$$

$$\leq \frac{1}{2} \sum_{d=1}^{|\mathcal{S}|^K} \left( \mathbb{E}_{\mathbb{P}(I^T | b_d = 1)}[\mathcal{N}_{g(d, b_d = 1, \bar{\theta})} \mathbb{I}(A_d)] + \mathbb{E}_{\mathbb{P}(I^T | b_d = 0)}[\mathcal{N}_{g(d, b_d = 0, \bar{\theta})} \mathbb{I}(A_d)] \right) + \frac{|\mathcal{S}|^K}{T^2}.$$

Now, on the event $A_d$, we know:

$$\mathcal{N}_{g(d, b_d = 1, \bar{\theta})} < \frac{2T}{|\mathcal{S}|^K}, \quad \mathcal{N}_{g(d, b_d = 0, \bar{\theta})} < \frac{2T}{|\mathcal{S}|^K}.$$

To control the average visit count under the mixture distribution $\mathbb{P}(I^T | \bar{\theta})$, we use:

$$\mathbb{P}(I^T | \bar{\theta}) = \tfrac{1}{2} \mathbb{P}(I^T | b_d = 1, \bar{\theta}) + \tfrac{1}{2} \mathbb{P}(I^T | b_d = 0, \bar{\theta}), \quad \text{where } g(d, b_d = 0, \bar{\theta}) = \text{trs}.$$

Thus, the total expectation becomes:

$$\mathbb{E}_{\mathbb{P}(I^T | b_d = 1)}[\mathcal{N}_{g(d, b_d = 1, \bar{\theta})}] + \mathbb{E}_{\mathbb{P}(I^T | b_d = 0)}[\mathcal{N}_{g(d, b_d = 0, \bar{\theta})}]$$

$$\leq \mathbb{E}_{\mathbb{P}(I^T | b_d = 1)}[\mathcal{N}_{g(d, b_d = 1, \bar{\theta})} \mathbb{I}(A_d)] + \mathbb{E}_{\mathbb{P}(I^T | b_d = 1)}[\mathcal{N}_{g(d, b_d = 0, \bar{\theta})} \mathbb{I}(A_d^c)]$$

$$+ \mathbb{E}_{\mathbb{P}(I^T | b_d = 0)}[\mathcal{N}_{g(d, b_d = 0, \bar{\theta})} \mathbb{I}(A_d)] + \mathbb{E}_{\mathbb{P}(I^T | b_d = 0)}[\mathcal{N}_{g(d, b_d = 0, \bar{\theta})} \mathbb{I}(A_d^c)].$$

We now apply the fact that $\mathbb{E}[\mathbb{I}(A_d^c)] \leq \frac{1}{T^3}$ and bound the remaining terms as follows:

$$\mathbb{E}_{\mathbb{P}(I^T | b_d = 1)}[\mathcal{N}_{g(d, b_d = 1, \bar{\theta})}] + \mathbb{E}_{\mathbb{P}(I^T | b_d = 0)}[\mathcal{N}_{g(d, b_d = 0, \bar{\theta})}]$$

$$\leq \mathbb{E}_{\mathbb{P}(I^T | b_d = 1)}[\mathcal{N}_{g(d, b_d = 1, \bar{\theta})} \mathbb{I}(A_d)]] + \mathbb{E}_{\mathbb{P}(I^T | b_d = 0)}[\mathcal{N}_{g(d, b_d = 0, \bar{\theta})} \mathbb{I}(A_d)] + \frac{1}{T^2}.$$

Next, using the upper bound $\mathcal{N}_{g(d, b_d = 1, \bar{\theta})} \mathbb{I}(A_d) \leq \frac{2T}{|\mathcal{S}|^K}$, we have:

$$\mathbb{E}_{\mathbb{P}(I^T | b_d = 1)}[\mathcal{N}_{g(d, b_d = 1, \bar{\theta})}] + \mathbb{E}_{\mathbb{P}(I^T | b_d = 0)}[\mathcal{N}_{g(d, b_d = 0, \bar{\theta})}]$$

$$\leq \mathbb{E}_{\mathbb{P}(I^T | b_d = 1)}[\mathcal{N}_{g(d, b_d = 1, \bar{\theta})} \mathbb{I}(A_d)] + \mathbb{E}_{\mathbb{P}(I^T | b_d = 0)}[\mathcal{N}_{g(d, b_d = 0, \bar{\theta})} \mathbb{I}(A_d)] + \frac{1}{T^2}$$

$$\leq \frac{2T}{|\mathcal{S}|^K} \mathbb{E}_{\mathbb{P}(I^T | b_d = 1)} \left[ \frac{|\mathcal{S}|^K}{2T} \mathcal{N}_{g(d, b_d = 1, \bar{\theta})} \mathbb{I}(A_d) \right]$$

$$+ \frac{2T}{|\mathcal{S}|^K} \mathbb{E}_{\mathbb{P}(I^T | b_d = 0)} \left[ \frac{|\mathcal{S}|^K}{2T} \mathcal{N}_{g(d, b_d = 0, \bar{\theta})} \mathbb{I}(A_d) \right] + \frac{1}{T^2}.$$

Finally, by Pinsker's inequality $|p - q|^2 \leq \mathrm{KL}(p \| q)$ and the fact $\mathcal{N}_{g(d, b_d = 1, \bar{\theta})} \mathbb{I}(A_d) \leq \frac{2T}{|\mathcal{S}|^K}$, we conclude:

$$\frac{2T}{|\mathcal{S}|^K} \left( \mathbb{E}_{\mathbb{P}(I^T | b_d = 1)} \left[ \frac{|\mathcal{S}|^K}{2T} \mathcal{N}_{g(d, b_d = 1, \bar{\theta})} \mathbb{I}(A_d) \right] - \mathbb{E}_{\mathbb{P}(I^T | \bar{\theta})} \left[ \frac{|\mathcal{S}|^K}{2T} \mathcal{N}_{g(d, b_d = 1, \bar{\theta})} \mathbb{I}(A_d) \right] \right)$$

$$\leq \frac{2T}{|\mathcal{S}|^K} \sqrt{\tfrac{1}{2} \mathrm{KL}(\mathbb{P}(I^T | b_d = 1) \| \mathbb{P}(I^T | \bar{\theta}))}.$$

Similarly, it holds that

$$\frac{2T}{|\mathcal{S}|^K} \left( \mathbb{E}_{\mathbb{P}(I^T | b_d = 0)} \left[ \frac{|\mathcal{S}|^K}{2T} \mathcal{N}_{g(d, b_d = 0, \bar{\theta})} \mathbb{I}(A_d) \right] - \mathbb{E}_{\mathbb{P}(I^T | \bar{\theta})} \left[ \frac{|\mathcal{S}|^K}{2T} \mathcal{N}_{g(d, b_d = 0, \bar{\theta})} \mathbb{I}(A_d) \right] \right)$$

$$\leq \frac{2T}{|\mathcal{S}|^K} \sqrt{\tfrac{1}{2} \mathrm{KL}(\mathbb{P}(I^T | b_d = 0) \| \mathbb{P}(I^T | \bar{\theta}))}.$$

Above all, we could conclude that

$$\frac{1}{2}\sum_{d=1}^{|\mathcal{S}|^K}\left(\mathbb{E}_{\mathbb{P}(I^T|b_d=1)}[\mathcal{N}_{g(d,b_d=1,\bar{\theta})}]+\mathbb{E}_{\mathbb{P}(I^T|b_d=0)}[\mathcal{N}_{g(d,b_d=0,\bar{\theta})}]\right)$$

$$\leq \frac{1}{2}\sum_{d=1}^{|\mathcal{S}|^K}\left(\mathbb{E}_{\mathbb{P}(I^T|b_d=1)}[\mathcal{N}_{g(d,b_d=1,\bar{\theta})}\mathbb{I}(A_d)]+\mathbb{E}_{\mathbb{P}(I^T|b_d=0)}[\mathcal{N}_{g(d,b_d=0,\bar{\theta})}\mathbb{I}(A_d)]\right)+\frac{|\mathcal{S}|^K}{T^2}$$

$$\leq \frac{1}{2}\sum_{d=1}^{|\mathcal{S}|^K}\mathbb{E}_{\mathbb{P}(I^T|\bar{\theta})}\left[\mathcal{N}_{g(d,b_d=1,\bar{\theta})}\mathbb{I}(A_d)+\mathcal{N}_{g(d,b_d=0,\bar{\theta})}\mathbb{I}(A_d)\right]+\frac{|\mathcal{S}|^K}{T^2}$$

$$+\frac{\sqrt{2}T}{|\mathcal{S}|^K}\sum_{d=1}^{|\mathcal{S}|^K}\sqrt{\text{KL}(\mathbb{P}(I^T\mid\bar{\theta})\|\mathbb{P}(I^T\mid b_d=1))}+\frac{\sqrt{2}T}{|\mathcal{S}|^K}\sum_{d=1}^{|\mathcal{S}|^K}\sqrt{\text{KL}(\mathbb{P}(I^T\mid\bar{\theta})\|\mathbb{P}(I^T\mid b_d=0))}$$

$$\leq \frac{1}{2}\mathbb{E}_{\mathbb{P}(I^T|\bar{\theta})}\left[\sum_{d=1}^{|\mathcal{S}|^K}\mathcal{N}_{g(d,b_d=1,\bar{\theta})}+\mathcal{N}_{g(d,b_d=0,\bar{\theta})}\right]+\frac{|\mathcal{S}|^K}{T^2}$$

$$+\frac{\sqrt{2}T}{|\mathcal{S}|^K}\sum_{d=1}^{|\mathcal{S}|^K}\left(\underbrace{\sqrt{\text{KL}(\mathbb{P}(I^T\mid\bar{\theta})\|\mathbb{P}(I^T\mid b_d=1))}}_{\text{term I}}+\underbrace{\sqrt{\text{KL}(\mathbb{P}(I^T\mid\bar{\theta})\|\mathbb{P}(I^T\mid b_d=0))}}_{\text{term II}}\right)$$

$$\leq \frac{T}{2}+\frac{|\mathcal{S}|^K}{T^2}+\frac{\sqrt{2}T}{|\mathcal{S}|^K}\sum_{d=1}^{|\mathcal{S}|^K}(\text{term I}+\text{term II}).$$

Next, consider the term $I$ and term $II$, we introduce the following lemma:

**Lemma B.5.**

$$KL(\mathbb{P}(I^T\mid\bar{\theta})\|\mathbb{P}(I^T\mid b_d=1,\bar{\theta}))=\mathbb{E}_{\mathbb{P}(I^T|\bar{\theta})}\left[\mathcal{N}_{g(d,b_d=1,\bar{\theta})}+\mathcal{N}_{g(d,b_d=0,\bar{\theta})}\right](\epsilon^2+\mathcal{O}(\epsilon^4)),$$

$$KL(\mathbb{P}(I^T\mid\bar{\theta})\|\mathbb{P}(I^T\mid b_d=0,\bar{\theta}))=\mathbb{E}_{\mathbb{P}(I^T|\bar{\theta})}\left[\mathcal{N}_{g(d,b_d=1,\bar{\theta})}+\mathcal{N}_{g(d,b_d=0,\bar{\theta})}\right](\epsilon^2+\mathcal{O}(\epsilon^4)).$$

Next, consider the following upper bound term that arises in the regret lower bound derivation:

$$\frac{1}{2}\sum_{d=1}^{|\mathcal{S}|^K}\mathbb{E}_{\mathbb{P}(I^T|b_d=1)}[\mathcal{N}_{g(d,b_d=1,\bar{\theta})}]+\mathbb{E}_{\mathbb{P}(I^T|b_d=0)}[\mathcal{N}_{g(d,b_d=0,\bar{\theta})}]. \tag{18}$$

Therefore, applying Lemma B.5

$$\frac{1}{2}\sum_{d=1}^{|\mathcal{S}|^K}\left(\mathbb{E}_{\mathbb{P}(I^T|b_d=1)}[\mathcal{N}_{g(d,b_d=1,\bar{\theta})}]+\mathbb{E}_{\mathbb{P}(I^T|b_d=0)}[\mathcal{N}_{g(d,b_d=0,\bar{\theta})}]\right)$$

$$\leq \frac{T}{2}+\frac{|\mathcal{S}|^K}{T^2}+\frac{\sqrt{2}T\epsilon}{|\mathcal{S}|^K}\sum_{d=1}^{|\mathcal{S}|^K}\sqrt{\mathbb{E}_{\mathbb{P}(I^T|\bar{\theta})}[\mathcal{N}_{g(d,b_d=1,\bar{\theta})}+\mathcal{N}_{g(d,b_d=0,\bar{\theta})}]}.$$

According to the fact that $\mathcal{N}_{g(d,b_d=1,\bar{\theta})}+\mathcal{N}_{g(d,b_d=0,\bar{\theta})}\leq T$:

$$\frac{1}{2}\sum_{d=1}^{|\mathcal{S}|^K}\left(\mathbb{E}_{\mathbb{P}(I^T|b_d=1)}[\mathcal{N}_{g(d,b_d=1,\bar{\theta})}]+\mathbb{E}_{\mathbb{P}(I^T|b_d=0)}[\mathcal{N}_{g(d,b_d=0,\bar{\theta})}]\right)$$

$$\leq \frac{T}{2}+\frac{|\mathcal{S}|^K}{T^2}+\frac{\sqrt{2}T\epsilon}{|\mathcal{S}|^K}\cdot|\mathcal{S}|^K\sqrt{T}=\frac{T}{2}+\frac{|\mathcal{S}|^K}{T^2}+\frac{\sqrt{2}T^{3/2}\epsilon}{|\mathcal{S}|^{K/2}}.$$

Next, the regret lower bound becomes:

$$\max_\theta \text{Reg}_T(\mathcal{M}_\theta) \geq \epsilon \left( T - \frac{1}{2} \sum_{d=1}^{|\mathcal{S}|^K} \left( \mathbb{E}_{\mathbb{P}(I^T|b_d=1)}[\mathcal{N}_{g(d,b_d=1,\bar\theta)}] + \mathbb{E}_{\mathbb{P}(I^T|b_d=0)}[\mathcal{N}_{g(d,b_d=0,\bar\theta)}] \right) \right)$$

$$\geq \epsilon \left( T - \frac{T}{2} - \frac{|\mathcal{S}|^K}{T^2} - \frac{\sqrt{2}T^{3/2}\epsilon}{|\mathcal{S}|^{K/2}} \right).$$

Let $\epsilon = \frac{|\mathcal{S}|^{K/2}}{\sqrt{T}} \cdot \frac{1}{\sqrt{32\sqrt{2}}}$, then if $\frac{|\mathcal{S}|^K}{T^2} < \frac{T}{8}$, the $\frac{|\mathcal{S}|^K}{T^2} \cdot \epsilon \leq \frac{T}{8}$.

$$\max_\theta \text{Reg}_T(\mathcal{M}_\theta) \geq \frac{1}{\sqrt{32}}|\mathcal{S}|^{K/2}\sqrt{T} \cdot \left( T - \frac{T}{2} - \frac{T}{8} - \frac{T}{8} \right) \leq \frac{|\mathcal{S}|^{K/2}\sqrt{T}}{32\sqrt{2}}.$$

## B.6 PROOF OF THEOREM 5.2:

To add the $|\mathcal{A}|^{K/2}$ term, we first introduce the following lemma:

**Lemma B.6.** *Let $\theta_0 = 0$, $\theta_1 = \epsilon$, and let $X^N = \{x_1, \ldots, x_N\}$.*

*Define the probability as follows:*

$$P_0(X^N) = (\tfrac{1}{2})^N,$$

$$P_1(X^N) = (\tfrac{1}{2} + \epsilon)^s (\tfrac{1}{2} - \epsilon)^{N-s}, \quad s = \sum_{i=1}^N x_i, \quad x_i \sim Bern(\tfrac{1}{2} + \theta_i\epsilon).$$

*Further, it holds that,*

$$P_M(x^N) = \frac{1}{2M}P_1(x^N) + \frac{2M-1}{2M}P_0(x^N),$$

$$P_{\mathcal{M}_0}(x^N) = P_0(x^N), \quad P_{\mathcal{M}_1}(x^N) = \frac{1}{M}P_1(x^N) + \frac{M-1}{M}P_0(x^N).$$

*Then the KL divergence can be bounded as follows:*

$$KL(P_M\|P_{\mathcal{M}_0}) \leq \frac{N\epsilon^2}{2M^2}, \quad KL(P_M\|P_{\mathcal{M}_1}) \leq \frac{N\epsilon^2}{2M^2}.$$

Next, define the policy class:

$$\Psi(\bar\theta) := \{\pi_\theta : \theta \in \bar\Theta_{b,\bar\theta}\},$$

Then, find a policy class cover:

$$\Omega(\bar\theta) := \{\Psi(\bar\theta_j)\}_{j=1}^{N_\varphi}, \text{ such that:}$$

(1) $|\Omega(\bar\theta)| = |\mathcal{A}|^{K-1}|\mathcal{A}|/2 = N_\varphi$;
(2) Any $\theta_1, \theta_2 \in \bar\Theta_b$ satisfy $\Psi(\theta_1) \cap \Psi(\theta_2) = \varnothing$ or $\Psi(\theta_1) = \Psi(\theta_2)$;
(3) $\Psi(\bar\theta) \in \Omega(\bar\theta)$.

Define the joint parameters set as:

$$\hat\Theta_{\Omega(\bar\theta)} := \bigcup_{\theta \in \Omega(\bar\theta)} \Theta_\theta, \quad \varphi \in [N_\varphi], \text{ s.t. exists a map } \varphi \mapsto \bar\theta_\varphi \in \Omega(\bar\theta).$$

For any $b \in \{0,1\}^{|\mathcal{S}|^K}$, to simplex, set $\theta_{\varphi,b} = \theta_{b,\bar\theta_\varphi}$. Then, the parameters set $\Theta$ is defined as:

$$\Theta := \{\theta_{b,\varphi} \mid \varphi \in [N_\varphi], \quad b \in \{0,1\}^{|\mathcal{S}|^K}\}.$$

Next, $\mu_{b,\varphi}$ is the uniform distribution over $[N_\varphi] \times \{0,1\}^{|\mathcal{S}|^K}$.

Then, it holds that

$$\max_{\theta \in \hat{\Theta}} \text{Reg}_T(\mathcal{M}_\theta) \geq \mathbb{E}_{\theta_{b,\varphi} \sim \mu_{b,\varphi}} \left[ \text{Reg}_T(\mathcal{M}_{\theta_{b,\varphi}}) \right].$$

Then, the regret satisfies the following equation:

$$
\begin{aligned}
\text{Reg}_T(\mathcal{M}_{\theta_{b,\psi}}) &= \mathbb{E} \left[ \text{Reg}_T(I^T, \mathcal{M}_{\theta_{b,\psi}}) \right] \\
&= \sum_{t=1}^{T} \sum_{\tau \in \Gamma(\pi)} \mathbb{E} \left[ \mathbb{I}(\tau_t = \tau, \ \ell_t = 1) \right] - \left( \tfrac{1}{2} - \epsilon \right) T \\
&= \epsilon \left( T - \sum_{\tau \in \Gamma(\pi_{\theta_{b,\psi}})} \mathbb{E}[\mathcal{N}_\tau] \right) \\
&= \epsilon \left( T - \sum_{d=1}^{|\mathcal{S}|^K} \mathbb{E} \left[ \mathcal{N}_{g(d,b_d,\theta_{b,\psi})} \right] \right).
\end{aligned}
$$

Next, we take the expectation of regret bound, and it holds that

$$
\begin{aligned}
&\mathbb{E}_{\theta_{b,\psi} \sim \mu_{b,\psi}} \left[ \text{Reg}_T(\mathcal{M}_{\theta_{b,\psi}}) \right] \\
&= \frac{1}{N_\varphi} \sum_{\psi=1}^{N_\varphi} \mathbb{E}_{b \sim \mu_b} \left[ \text{Reg}_T(\mathcal{M}_{\theta_{b,\psi}}) \right] \\
&= \frac{1}{N_\varphi} \sum_{\psi=1}^{N_\varphi} \mathbb{E}_{b \sim \mu_b} \left[ \epsilon \left( T - \sum_{d=1}^{|\mathcal{S}|^K} \mathbb{E}_{\mathcal{M}_{b,\psi}} \left[ \mathcal{N}_{g(d,b_d,\theta_{b,\psi})} \right] \right) \right] \\
&= \epsilon T \left( 1 - \frac{1}{2T} \sum_{d=1}^{|\mathcal{S}|^K} \left( \frac{1}{N_\varphi} \sum_{\psi=1}^{N_\varphi} \left( \mathbb{E}_{\mathcal{P}(I^T | b_d = 1, \theta_{b,\psi})} \left[ \mathcal{N}_{g(d,b_d=1,\theta_{b,\psi})} \right] \right. \right. \right. \\
&\qquad \left. \left. \left. + \mathbb{E}_{\mathcal{P}(I^T | b_d = 0, \theta_{b,\psi})} \left[ \mathcal{N}_{g(d,b_d=0,\theta_{b,\psi})} \right] \right) \right) \right).
\end{aligned}
$$

Next, $\mu_{\theta,b}$ uniform distribution over $[N_\varphi] \times \{0,1\}^{|\mathcal{S}|^K}$.

Then, it holds that

$$\max_{\theta \in \hat{\Theta}} \text{Reg}_T(\mathcal{M}_\theta) \geq \mathbb{E}_{\theta_\psi, b} \left[ \text{Reg}_T(\mathcal{M}_{\theta_\psi, b}) \right].$$

Further, we can rewrite the regret as:

$$
\begin{aligned}
\text{Reg}_T(\mathcal{M}_{\theta_\psi, b}) &= \mathbb{E} \left[ \text{Reg}_T(I^T, \mathcal{M}_{\theta_\psi, b}) \right] \\
&= \sum_{t=1}^{T} \sum_{\tau \in \Gamma_{\theta_\psi}} \mathbb{E} \left[ \mathbb{I}(\tau_t = \tau, \ell_t = 1) \right] - \left( \tfrac{1}{2} + \epsilon \right) T \\
&= \epsilon \left( T - \sum_{\tau \in \Gamma(\pi_{\theta_\psi, b})} \mathbb{E}[\mathcal{N}_\tau] \right) \\
&= \epsilon \left( T - \sum_{d=1}^{|\mathcal{S}|^K} \mathbb{E}[\mathcal{N}_g(d, b_d, \theta_{b,\psi})] \right).
\end{aligned}
$$

Next, to convenience, we rewrite $\mathcal{N}_g(d, b_d, \theta_{b,\psi}) = \mathcal{N}_g(d, b_d, \theta_\psi)$. Then, it holds that

$$
\mathbb{E}_{\theta_\psi, b \sim \mu_{\theta,b}} \left[ \mathrm{Reg}_T(\mathcal{M}_{\theta_\psi, b}) \right] = \frac{1}{N_\varphi} \sum_{\psi=1}^{N_\varphi} \mathbb{E}_{b \sim \mu_b} \left[ \mathrm{Reg}_T(\mathcal{M}_{\theta_\psi, b}) \right]
$$

$$
= \frac{1}{N_\varphi} \sum_{\psi=1}^{N_\varphi} \mathbb{E}_{b \sim \mu_b} \left[ \epsilon \left( T - \sum_{d=1}^{|\mathcal{S}|^K} \mathbb{E}_{\mathbb{P}_{\mathcal{M}_{\theta_\psi, b}}} [\mathcal{N}_g(d, b_d, \theta_\psi)] \right) \right]
$$

$$
= \epsilon T \left( 1 - \frac{1}{2T} \sum_{d=1}^{|\mathcal{S}|^K} \left( \frac{1}{N_\varphi} \sum_{\psi=1}^{N_\varphi} \mathbb{E}_{\mathbb{P}(I^T | b_d=1, \theta_\psi)} [\mathcal{N}_g(d, b_d = 1, \theta_\psi)] \right. \right.
$$

$$
\left. \left. + \mathbb{E}_{\mathbb{P}(I^T | b_d=0, \theta_\psi)} [\mathcal{N}_g(d, b_d = 0, \theta_\psi)] \right) \right).
$$

Set:

$$
q = \frac{1}{N_\varphi} \sum_{\psi=1}^{N_\varphi} P(\cdot \mid \theta_\varphi), \quad q(\cdot \mid b_d = 1, \theta_\psi) = \frac{1}{N_\varphi} \sum_{\psi=1}^{N_\varphi} P(\cdot \mid \theta_\psi) + \frac{1}{N_\varphi} P(\cdot \mid b_d = 1, \theta_\beta) \quad (\text{resp. } bd = 0).
$$

Then, the regret lower bound can be bounded as:

$$
\max_\theta \mathrm{Reg}_T(\mathcal{M}_\theta)
$$

$$
\geq \epsilon T \left( 1 - \frac{1}{2T} \sum_{d=1}^{|\mathcal{S}|^K} \sum_{\psi=1}^{N_\varphi} \left( \frac{1}{N_\varphi} \sum_{\psi=1}^{N_\varphi} \mathbb{E}_{P(\cdot | b_d=1, \theta_\psi)} [\mathbb{I}_{\mathcal{N}_g(b_d=1, \theta_\psi)}] + \frac{1}{N_\varphi} \sum_{\psi \neq \varphi} \mathbb{E}_{P(\cdot | \theta_\varphi)} [\mathbb{I}_{\mathcal{N}_g(d, b_d=1, \theta_\psi)}] \right) \right)
$$

$$
\geq \epsilon T \left( 1 - \frac{1}{2T} \sum_{d=1}^{|\mathcal{S}|^K} \sum_{\psi=1}^{N_\varphi} \underbrace{\left( \mathbb{E}_{q(\cdot | b_d=1, \theta_\psi)} [\mathbb{I}_{\mathcal{N}_g(b_d=1, \theta_\psi)}] + \mathbb{E}_{q_c(\cdot | b_d=0, \theta_\psi)} [\mathbb{I}_{\mathcal{N}_g(b_d=0, \theta_\psi)}] \right)}_{\text{term I}} \right)
$$

**Term I Estimate**. From the Lemma B.4 and proof in the previous section, we could bound:

$$
\text{Term I} \leq \mathbb{E}_q [\mathbb{I}_{\mathcal{N}_g(d, b_d=1, \theta_\psi)} + \mathbb{I}_{\mathcal{N}_g(d, b_d=0, \theta_\psi)}]
$$

$$
+ \frac{2T}{|\mathcal{S}|^K} \left( \sqrt{\frac{1}{2} \mathrm{KL}(q \| q(\cdot \mid d, b_d = 1, \theta_\psi))} + \sqrt{\frac{1}{2} \mathrm{KL}(q \| q(\cdot \mid d, b_d = 0, \theta_\psi))} \right) + \frac{1}{T^3}.
$$

Then summing over all $d, \psi$:

$$
\sum_{d=1}^{|\mathcal{S}|^K} \sum_{\psi=1}^{N_\varphi} \text{Term I} \leq \mathbb{E}_q \left[ \sum_{d=1}^{|\mathcal{S}|^K} \sum_{\psi=1}^{N_\varphi} (\mathbb{I}_{\mathcal{N}_g(d, b_d=1, \theta_\psi)} + \mathbb{I}_{\mathcal{N}_g(d, b_d=0, \theta_\psi)}) \right] + \frac{|\mathcal{S}|^K N_\varphi}{T^2}
$$

$$
\leq T + \frac{|\mathcal{S}|^K N_\varphi}{T^2} + \frac{T}{|\mathcal{S}|^K} \cdot \frac{\epsilon}{N_\varphi} \cdot \sqrt{\mathbb{E}_q \left[ \mathbb{I}_{\mathcal{N}_g(d, b_d=1, \theta_\psi)} + \mathbb{I}_{\mathcal{N}_g(d, b_d=0, \theta_\psi)} \right]} \cdot \sqrt{T}
$$

Next, we define the joint distribution as follows:

$$
\max_{\theta \in \Theta} \mathrm{Reg}_T(\mathcal{M}_\theta) \geq \epsilon T \left( 1 - \frac{1}{2T} \left( T + \frac{|\mathcal{S}|^K N_\varphi}{T^2} + \frac{\epsilon T \cdot \sqrt{T}}{2\sqrt{|\mathcal{S}|^K N_\varphi}} \right) \right)
$$

$$
= \epsilon T \left( \tfrac{1}{2} - \frac{|\mathcal{S}|^K N_\varphi}{2T^3} - \frac{\epsilon \sqrt{T}}{\sqrt{|\mathcal{S}|^K N_\varphi}} \right).
$$

Here, set $\epsilon = \frac{\sqrt{2|\mathcal{S}|^K N_\varphi}}{4\sqrt{T}} = \frac{\sqrt{2|\mathcal{S}|^K |\mathcal{S}|^K}}{4\sqrt{T}}$, if $T \geq (|\mathcal{S}||\mathcal{A}|)^K \log T$, then

$$
\max_{\theta \in \hat{\Theta}} \mathrm{Reg}_T(\mathcal{M}_\theta) \geq \sqrt{|\mathcal{S}|^K N_\varphi T} \cdot \frac{\sqrt{2}}{8} \left( \tfrac{1}{2} - \tfrac{1}{8} - \tfrac{1}{4} \right) \geq \frac{\sqrt{(|\mathcal{S}||\mathcal{A}|)^K T}}{128}.
$$

If $T \leq 4(|\mathcal{S}||\mathcal{A}|)^K \log T$, we frozen some states, i.e. $\mathcal{N}_{\mathcal{S}} = \coprod_{k=0}^{K} |\mathcal{S}^k|_{active}$, and $2\mathcal{N}_{\mathcal{S}}|\mathcal{A}|^K \log T \geq T \geq 4\mathcal{N}_{\mathcal{S}}|\mathcal{A}|^K \log T$. Then, with similar proof process,

$$\max_{\theta \in \hat{\Theta}} \text{Reg}_T(\mathcal{M}_\theta) \geq \frac{\sqrt{\mathcal{N}_{\mathcal{S}}|\mathcal{A}|^K T}}{128} \geq \frac{2\mathcal{N}_{\mathcal{S}}|\mathcal{A}|^K}{128 \log T} \geq \frac{T}{128 \log^2 T}.$$

The proof completed.

## C  PROOF OF LEMMAS

*Proof of Lemma A.3.* Here, we provide the proof of Lemma A.3. We set $\beta := \lambda\gamma$ and $\lambda = 2$. Then, we do error decomposition as follows:

$$\widehat{\mathcal{L}}_t(\tau) = \frac{\ell_t(\tau) \cdot \mathbb{I}_{\tau=\tau_t}}{\Pi_t(\tau) + \gamma} \leq \frac{\ell_t(\tau) \cdot \mathbb{I}_{\tau=\tau_t}}{\Pi_t(\tau) + \gamma\ell_t(\tau)} \leq \frac{\mathbb{I}_{\tau=\tau_t}}{\beta} \cdot \frac{\frac{\beta\ell_t(\tau)}{\Pi_t(\tau)}}{1 + \frac{\beta\ell_t(\tau)}{2\Pi_t(\tau)}}.$$

From the inequality $\frac{z}{1+z/2} \leq \log(1+z)$ for all $z \geq 0$, we have:

$$\widehat{\mathcal{L}}_t(\tau) \leq \frac{1}{\beta} \log\left(1 + \frac{\beta \cdot \ell_t(\tau) \cdot \mathbb{I}_{\tau=\tau_t}}{\Pi_t(\tau)}\right)$$

Then, to simplex the presentation, we rewrite $\mathbb{E}_t := \mathbb{E}_{\tau_t \sim \{P_t, \pi_t\}}$. Combining with the inequality $z_1 \log(1 + z_2) \leq \log((1 + z_1 z_2))$ for $z_1 \leq 1$, we can get:

$$\mathbb{E}_t\left[\exp\left(\sum_{\tau \in \mathcal{C}_\tau} \alpha_t(\tau)\widehat{\mathcal{L}}_t(\tau)\right)\right] \leq \mathbb{E}_t\left[\exp\left(\sum_{\tau \in \mathcal{C}_\tau} \frac{\alpha_t(\tau)}{\beta} \cdot \log\left(1 + \frac{\beta \cdot \ell_t(\tau) \cdot \mathbb{I}_{\tau=\tau_t}}{\Pi_t(\tau)}\right)\right)\right]$$

$$\overset{(a)}{\leq} \mathbb{E}_t\left[\prod_\tau \left(1 + \frac{\alpha_t(\tau) \cdot \ell_t(\tau) \cdot \mathbb{I}_{\tau=\tau_t}}{\Pi_t(\tau)}\right)\right]$$

$$\overset{(b)}{=} \mathbb{E}_t\left[1 + \sum_{\tau \in \mathcal{C}_\tau} \frac{\alpha_t(\tau) \cdot \ell_t(\tau) \cdot \mathbb{I}_{\tau=\tau_t}}{\Pi_t(\tau)}\right]$$

$$= 1 + \mathbb{E}\left[\sum_{\tau \in \mathcal{C}_\tau} \frac{\alpha_t(\tau) \cdot \ell_t(\tau) \cdot \mathbb{I}_{\tau=\tau_t}}{\Pi_t(\tau)}\right]$$

$$\overset{(c)}{=} 1 + \sum_{\tau \in \mathcal{C}_\tau} \alpha_t(\tau)\widetilde{\mathcal{L}}_t(\tau)\mathbb{I}_{\tau=\tau_t}$$

$$\leq \exp\left(\sum_{\tau \in \mathcal{C}_\tau} \alpha_t(\tau)\mathcal{L}_t(\tau)\right), \qquad (19)$$

where $(a)$ from the constrict that $\frac{\alpha_t(\tau)}{2\gamma} < 1$ and $(b)$ follows from the fact that

$$\mathbb{I}_{\tau'=\tau_t}\mathbb{I}_{\tau=\tau_t} = 0; \qquad\qquad \mathbb{I}_{\tau=\tau_t}\mathbb{I}_{\tau=\tau_t} = \mathbb{I}_{\tau=\tau_t}.$$

The $(c)$ follows from the Lemma A.1.

Then, we can further get

$$\mathbb{P}\left(\sum_{t=1}^{T}\sum_{\tau\in\mathcal{C}_{\tau}}\alpha_t(\tau)\left(\widehat{\mathcal{L}}_t(\tau)-\mathcal{L}_t(\tau)\right)>\log\left(\frac{1}{\delta}\right)\right)$$

$$\leq\delta\mathbb{E}\left[\exp\left(\sum_{t=1}^{T}\sum_{\tau\in\mathcal{C}_{\tau}}\alpha_t(\tau)\left(\widehat{\mathcal{L}}_t(\tau)-\mathcal{L}_t(\tau)\right)\right)\right]$$

$$=\delta\mathbb{E}\left[\exp\left(\sum_{t=1}^{T-1}\sum_{\tau\in\mathcal{C}_{\tau}}\alpha_t(\tau)\left(\widehat{\mathcal{L}}_t(\tau)-\mathcal{L}_t(\tau)\right)\right)\mathbb{E}\left[\exp\left(\sum_{\tau\in\mathcal{C}_{\tau}}\alpha_t(\tau)\left(\widehat{\mathcal{L}}_T(\tau)-\mathcal{L}_T(\tau)\right)\right)\right]\right]$$

$$\overset{(a)}{\leq}\delta\mathbb{E}\left[\exp\left(\sum_{t=1}^{T-1}\sum_{\tau\in\mathcal{C}_{\tau}}\alpha_t(\tau)\left(\widehat{\mathcal{L}}_t(\tau)-\mathcal{L}_t(\tau)\right)\right)\right]$$

$$\leq\delta...\leq\delta,$$

where $(a)$ follows from the Eq. 19.

Therefore, we can get that

$$\sum_{t=1}^{T}\sum_{\tau\in\mathcal{C}_{\tau}}\alpha_t(\tau)\left(\widehat{\mathcal{L}}_t(\tau)-\mathcal{L}_t(\tau)\right)\leq\log\left(\frac{1}{\delta}\right).$$

Proof completed here. $\square$

*Proof of Lemma A.4.* For any two policy $\pi,\pi'\in\mathcal{C}_{\pi}$ and any $\tau\in\mathcal{C}_{\tau}$, it holds that

$$\Pi_{\pi}(\tau)=\prod_{k=0}^{K-1}\pi(a^k\mid h^k),\Pi_{\pi'}(\tau)=\prod_{k=0}^{K-1}\pi'(a^k\mid h^k).$$

For $\bar{\Pi}_{\pi}=\lambda\Pi_{\pi}+(1-\lambda)\Pi_{\pi'}$, where $\lambda\in[0,1]$, we can get the $\bar{\pi}$ from $\bar{\Pi}_{\pi}$. For convenience, we set $\Pi(h^k,a^k)=\prod_{j=0}^{k}\pi(a^j|h^j)$ and $\Pi(h^k)=\sum_{a^k}\Pi(h^k,a^k)$. Clearly, given the vector $\Pi$, we can get the value of $\Pi(h^k,a^k)$ and $\Pi(h^k)$ easily by the iteration $\Pi(h^k,a^k)=\Pi(h^{k+1}),h^k,a^k\subset h^{k+1}$ and $\Pi(h^k)=\sum_{a^k}\Pi(h^k,a^k)$.

Next, set $\bar{\pi}$ is the policy corresponding to the vector $\bar{\Pi}$, we next prove that $\bar{\pi}(a^k|h^k)=\frac{\lambda\Pi_{\pi}(h^k,a^k)+(1-\lambda)\Pi_{\pi'}(h^k,a^k)}{\lambda\Pi_{\pi}(h^k)+(1-\lambda)\Pi_{\pi'}(h^k)}$ and $\bar{\pi}(a^k|h^k)\in\mathcal{C}_{\pi}$.

Firstly, it holds that

$$\sum_{a^k}\bar{\pi}(a^k|h^k)=\sum_{a^k}\frac{\lambda\Pi_{\pi}(h^k,a^k)+(1-\lambda)\Pi_{\pi'}(h^k,a^k)}{\lambda\Pi_{\pi}(h^k)+(1-\lambda)\Pi_{\pi'}(h^k)}$$

$$=\frac{\lambda\sum_{a^k}\Pi_{\pi}(h^k,a^k)+(1-\lambda)\sum_{a^k}\Pi_{\pi'}(h^k,a^k)}{\lambda\Pi_{\pi}(h^k)+(1-\lambda)\Pi_{\pi'}(h^k)}$$

$$=\frac{\lambda\Pi_{\pi}(h^k)+(1-\lambda)\Pi_{\pi'}(h^k)}{\lambda\Pi_{\pi}(h^k)+(1-\lambda)\Pi_{\pi'}(h^k)}=1.$$

Therefore, the $\bar{\pi}$ is a policy.

Moreover, since policy $\pi$ and $\pi'$ are $\epsilon$-greedy policy, i.e. $\pi(a^k|h^k),\pi'(a^k|h^k)\geq\epsilon$, it holds that

$$\pi(a^k|h^k)=\sum_{a^k}\frac{\lambda\Pi_{\pi}(h^k,a^k)+(1-\lambda)\Pi_{\pi'}(h^k,a^k)}{\lambda\Pi_{\pi}(h^k)+(1-\lambda)\Pi_{\pi'}(h^k)}\geq\min\left\{\frac{\Pi_{\pi}(h^k,a^k)}{\Pi_{\pi}(h^k)},\frac{\Pi_{\pi'}(h^k,a^k)}{\Pi_{\pi'}(h^k)}\right\}$$

$$=\min\{\pi(a^k|h^k),\pi'(a^k|h^k)\}\geq\epsilon.$$

Therefore, the $\bar{\pi}$ is an $\epsilon$-greedy policy. Above all, the policy $\bar{\pi}\subset\mathcal{C}_{\pi}$.

Besides, we have that $\Pi(h^{k-1}, a^k) = \Pi(h^k)$ for $\{h^{k-1}, a^k\} \subset h^k$. Then, if $\{h^k, a^k\} \subset \tau$ for $k = 0, ..., K-1, \tau \in \mathcal{C}_\tau$, we can get

$$\prod_{k=0}^{K-1} \bar{\pi}(a^k | h^k) = \prod_{k=0}^{K-1} \frac{\lambda \Pi_\pi(h^k, a^k) + (1-\lambda)\Pi_{\pi'}(h^k, a^k)}{\lambda \Pi_\pi(h^k) + (1-\lambda)\Pi_{\pi'}(h^k)}$$

$$= (\lambda \Pi_\pi(h^0, a^0) + (1-\lambda)\Pi_{\pi'}(h^0, a^0)) \prod_{k=1}^{K-1} \frac{\lambda \Pi_\pi(h^k, a^k) + (1-\lambda)\Pi_{\pi'}(h^k, a^k)}{\lambda \Pi_\pi(h^{k-1}, a^{k-1}) + (1-\lambda)\Pi_{\pi'}(h^{k-1}, a^{k-1})}$$

$$= (\lambda \Pi_\pi(h^{K-1}, a^{K-1}) + (1-\lambda)\Pi_{\pi'}(h^{K-1}, a^{K-1})) = \lambda \Pi_\pi(\tau) + (1-\lambda)\Pi_{\pi'}(\tau).$$

Policy $\bar{\pi}$ is the policy corresponding to the vector $\bar{\Pi}$.

Above all, proof completed here. $\qquad\square$

**Lemma C.1.** *Define the averaged distribution*

$$\tilde{P}_{\mathcal{M}_\theta}(\tau) := \mathbb{E}_{P \sim \mathcal{M}_\theta}\left[\tilde{P}(\tau)\right] \text{ and } \tilde{P}_{\mathcal{M}_\theta}(\tau, r) := \mathbb{E}_{P \sim \mathcal{M}_\theta}\left[\tilde{P}(\tau)P(r \mid \tau)\right],$$

*where $\tilde{P}(\tau) = \prod_{k=0}^{K-1} P(s^{k+1} \mid s^k, a^k)$, $s^k, a^k, s^{k+1} \in \tau$.*

*Then,*

$$\tilde{P}_{\mathcal{M}_\theta}(\tau) = \frac{1}{|\mathcal{S}|^K};$$

$$\tilde{P}_{\mathcal{M}_\theta}(\tau, r = 1) = \begin{cases} \frac{1}{|\mathcal{S}|^K}\left(\frac{1}{2} + \epsilon\right), & \text{if } \tau \in \Gamma(\pi_\theta) \\ \frac{1}{|\mathcal{S}|^K} \cdot \frac{1}{2}, & \text{otherwise.} \end{cases}$$

*Proof of Lemma C.1.* Next, consider the hard MDPs described in the figure/definition.

If $\tau' \in \Gamma(\pi_\theta)$, then

$$\tilde{P}_{\mathcal{M}_\theta}(\tau) = \frac{1}{|\mathcal{S}|^K} \sum_{\tau' \in \Gamma(\pi_\theta)} P_\tau(\tau') = \frac{1}{|\mathcal{S}|^K},$$

$$\tilde{P}_{\mathcal{M}_\theta}(\tau, r = 1) = \frac{1}{|\mathcal{S}|^K}\left(\frac{1}{2} + \epsilon\right).$$

Define the set $\Gamma^{K-1}(\pi_\theta)$, i.e.,

$$\Gamma^K(\pi_\theta) \triangleq \left\{\tau : \prod_{k=0}^{K-1} \pi_\theta(a^k \mid h^k) = 1\right\},$$

$$\Gamma^{K-1}(\pi_\theta) \triangleq \left\{\tau : \prod_{k=0}^{K-2} \pi_\theta(a^k \mid h^k) = 1, \pi_\theta(a^{K-1} \mid h^{K-1}) = 0\right\},$$

$$\Gamma^k(\pi_\theta) \triangleq \left\{\tau : \prod_{k=0}^{k-1} \pi_\theta(a^k \mid h^k) = 1, \prod_{k=0}^{K-1} \pi_\theta(a^{K-1} \mid h^{K-1}) = 0\right\}$$

**Remark 7.** *A trajectory in this set means: at the $K$-th step, the action is not optimal.*

If trajectory $\tau' \in \Gamma^{K-1}(\pi_\theta)$, then

$$\tilde{P}_{\mathcal{M}_\theta}(\tau) = \frac{1}{|\mathcal{S}|^K} \sum_{\tau' \in \Gamma(\pi_\theta)} P_\tau(\tau')$$

$$= \frac{1}{|\mathcal{S}|^K} \sum_{\tau' \in \Gamma(\pi_\theta)} \mathbb{I}(\tau' \neq \tau, h^{K-1} \subset \tau \text{ and } h^{K-1} \subset \tau')$$

$$= \frac{1}{|\mathcal{S}|^K} \cdot \frac{|\mathcal{S}| - 1}{|\mathcal{S}| - 1} = \frac{1}{|\mathcal{S}|^K}.$$

Here, the event $\{\tau' \neq \tau, h^{K-1} \subset \tau, h^{K-1} \subset \tau'\}$ indicates that the current trajectory is consistent with the past optimal actions except the final step.

We now evaluate the probability of a suboptimal trajectory $\tau' \notin \Gamma(\pi_\theta)$ under the averaged distribution.

$$\tilde{P}_{\mathcal{M}_\theta}(\tau') = \frac{1}{|\mathcal{S}|^K} \sum_{\tau \in \Gamma(\pi_\theta)} P_\tau(\tau')$$

$$= \frac{1}{|\mathcal{S}|^K} \cdot \frac{(|\mathcal{S}| - 1)^{K-k}}{(|\mathcal{S}| - 1)^{K-k}} = \frac{1}{|\mathcal{S}|^K}.$$

The individual term inside the sum is

$$P_\tau(\tau') = \frac{1}{(|\mathcal{S}| - 1)^{K-k}} \cdot \mathbb{I}(\tau' \text{ is available in } P_\tau),$$

where availability means the trajectory is reachable under $P_\tau$.

The total number of such trajectories satisfies

$$\sum_{\tau \in \Gamma(\pi_\theta)} \mathbb{I}(\tau' \text{ is available in } P_\tau) = (|\mathcal{S}| - 1)^{K-k}.$$

Thus, the expected joint distribution for a suboptimal trajectory is

$$\tilde{P}_{\mathcal{M}_\theta}(\tau) = \frac{1}{|\mathcal{S}|^K} \text{ and } \tilde{P}_{\mathcal{M}_\theta}(\tau, r = 1) = \frac{1}{2|\mathcal{S}|^K} \text{ if } \tau \notin \Gamma(\pi_\theta).$$

Above all, the proof ends here. $\qquad\square$

*Proof of Lemma B.4* . Firstly, it holds that given any $I^T$,

$$\mathcal{N}_g(d, \bar{\theta}) = \sum_{t=1}^T \mathbb{I}_{\tau_t = g(d, b_d = 1, \bar{\theta})} + \sum_{t=1}^T \mathbb{I}_{\tau_t = g(d, b_d = 0, \bar{\theta})}.$$

From the definition of $g(d, b_d, \bar{\theta})$, both forms are instances of $\Gamma(\mathcal{T}_S(d), \bar{\theta})$.

Set

$$\mathcal{N}_{\text{trs}} := \sum_{t=1}^T \mathbb{I}(trs(\tau_t) = \text{trs}).$$

We observe that

$$\{\tau : \tau = g(d, b_d, \bar{\theta})\} \subseteq \{\tau : \text{trs}(\tau) = \text{trs}(d)\}.$$

Therefore, we need to prove that

$$\mathbb{P}(\mathcal{N}_{\text{trs}} > \tfrac{2T}{|\mathcal{S}|^K}) < \tfrac{1}{T^3}, \quad \text{for } T > 2|\mathcal{S}|^K \sqrt{T \log T}.$$

Next, we prove:

(1) $\mathbb{P}(\mathcal{N}_{\text{trs}}, \mathcal{N}_{\text{trs}'}) = \mathbb{P}(\mathcal{N}_{\text{trs}}, \mathcal{N}_{\text{trs}'})$, i.e., trs i.i.d.

(2) $\mathbb{P}(trs(\tau) = \text{trs}) = \frac{1}{|\mathcal{S}|^K}$.

We compute:

$$\mathbb{E}[\mathcal{N}_{\text{trs}}] = \mathbb{E}\left[\sum_{t=1}^{T} \mathbb{I}(trs(\tau_t) = \text{trs})\right]$$

$$= \sum_{t=1}^{T} \mathbb{E}\left[\mathbb{I}(trs(\tau_t) = \text{trs})\right]$$

$$= \sum_{t=1}^{T} \sum_{\tau_t \in \mathcal{C}_\tau, trs(\tau_t)=\text{trs}} \mathbb{P}_{I^T \sim \mathcal{M}_\theta}(\tau_t)$$

$$= \sum_{t=1}^{T} \sum_{a^k \in \mathcal{A}^k} \left(\prod_{k=0}^{K-1} \text{Alg}(a^k \mid \nu_t^k)\right) \tilde{\mathbb{P}}_{\mathcal{M}_\theta}(\tau_t)$$

$$= \sum_{t=1}^{T} \tilde{\mathbb{P}}_{\mathcal{M}_\theta}(\tau_t) = \frac{T}{|\mathcal{S}|^K}.$$

Thus, $\mathcal{N}_{\text{trs}} \sim \text{Binomial}(\frac{1}{|\mathcal{S}|^K}, T)$. Therefore,

$$\text{Var}(\mathcal{N}_{\text{trs}}) = T\left(\frac{1}{|\mathcal{S}|^K}\right)\left(1 - \frac{1}{|\mathcal{S}|^K}\right) \le \frac{\sqrt{T}}{|\mathcal{S}|^{K/2}}.$$

Using Bernstein or Chernoff bounds,

$$\mathbb{P}\left(\mathcal{N}_{\text{trs}} - \frac{T}{|\mathcal{S}|^K} > \frac{T}{|\mathcal{S}|^K}\right) \le \mathbb{P}\left(\mathcal{N}_{\text{trs}} > \frac{2T}{|\mathcal{S}|^K}\right) \le \exp\left(-\frac{T}{|\mathcal{S}|^K} \cdot \frac{1}{3}\right) \le \exp\left(-3(\log T)\right) \le \frac{1}{T^3}.$$

$\square$

