# OpenReview forum: "Minimax Optimal Adversarial Reinforcement Learning"
_ICLR.cc/2026/Conference — ICLR 2026 Poster_

### Official Review · Reviewer_jRLS · 2025-10-19

**Soundness:** 3
**Presentation:** 3
**Contribution:** 3
**Rating:** 6
**Confidence:** 4

**Summary:**

This paper addresses Adversarial Reinforcement Learning (RL) where both the transition kernel and the reward function exhibit adversarial dynamics across different episodes. To tackle this, the authors propose AD-FTRL, a novel algorithm based on occupancy measure analysis. This method achieves a regret bound proportional to the square root of the number of episodes ($\sqrt{T}$). Furthermore, the authors include a counterexample to suggest the optimality of their proposed framework.

**Strengths:**

1. This paper tackles the challenging problem of fully adversarial Reinforcement Learning (RL), where both the transition kernel and the reward function exhibit adversarial dynamics across episodes. The proposed framework comes with a strong theoretical guarantee of $\mathcal{O}(\sqrt{T})$ regret.

2. The authors provide counterexamples that demonstrate the necessity of non-Markovian policies and further illustrate the optimality of their proposed algorithm.

**Weaknesses:**

1. Even though the lower bound suggests that the regret is optimal, the regret bound has an exponential dependency on the size of the state-action space ($|S||A|$). Consequently, the regret bound is non-trivial only when the number of episodes ($T$) is sufficiently large (specifically, $T > |S||A|^k$). This exponential dependence significantly limits the framework's practical application to large-scale problems.

2. Some claims regarding related work are improper. Specifically, concerning the Upper Bound Analysis, it is crucial to note that prior analyses for adversarial reinforcement learning or bandits operate in three distinct settings, each with a different regret definition. It is unfair to make a comparison without clearly stating these differences.

(a). Fully Adversarial RL: This work falls into this setting (discussed in Lines 118-119), where there is no limitation on the adversarial dynamics, and regret is calculated against the best fixed policy across all episodes. Prior work in this setting did not consider a dynamic transition kernel.

(b). Adversarial Corruption: This setting (Lines 128-129) focuses on corruption where a base reward function and transition kernel exist, and only a small corruption is added. Regret is calculated against the optimal policy in the base model and is efficient only when the total corruption level ($C$) is sublinear in the number of episodes.

(c). Non-Stationary Environments: In this setting (Lines 134-135), the regret is measured against the optimal action in each episode (not fixed across episodes), making the guarantee strictly stronger than the guarantee in this paper. However, this comes at the cost of having a limitation on the cumulative dynamic.

Overall, it is necessary to clarify the different settings to accurately establish the contribution of this work.

3. The claim in Remark 1 is incorrect. There still exists a trivial method that treats each Markovian policy $\mathcal{S} \times \mathcal{H} \to \mathcal{A}$ as a distinct bandit arm or agent, which yields $|\mathcal{A}|^{|\mathcal{S}|H}$ different arms. Even though the result from this trivial method is non-optimal, the authors still need to clarify the mapping.

**Questions:**

1. The preliminary discussion ("WARM-UP") primarily focuses on the adversarial transition kernel. While it is true that learning transitions in a stochastic environment is often harder than learning the reward function, the analysis should address the additional challenges and complexities that arise when combining adversarial rewards and adversarial transition kernels simultaneously. A deeper discussion on this interaction is warranted.

2. The current framework assumes the action and state sets are non-overlapping across different stages, and, more specifically, assumes uniform sizes such as $|A^k|=|A|$ and $|S^k|=|S|$. What is the regret bound without these simplifying assumptions? The paper should discuss the impact of varying state and action space sizes across stages.

3. Several key related works on Adversarial Reinforcement Learning are missing and should be included for proper context and comparison:

(a): Fully Adversarial RL (Under Function Approximation): The analysis of fully adversarial RL extends beyond tabular settings when incorporating function approximation:

[1] "Provably Efficient Exploration in Policy Optimization" (ICML 2020) considered the adversarial reward function under linear function approximation.

[2] "Near-optimal policy optimization algorithms for learning adversarial linear mixture mdps" (AISTATS 2022) later extended the results to achieve optimal guarantees within the linear function approximation setting.

(b): Adversarial Corruption (General Function Approximation): In the setting of adversarial corruption, work has also addressed robustness with complex function classes:

[3] "Towards robust model-based reinforcement learning against adversarial corruption" (ICML 2024) studied adversarial attacks on the transition kernel with general function approximation, achieving a near-optimal regret guarantee under the adversarial attack model.

---

> ### Author Response · Authors · 2025-11-21
>
> ***Weaknesses 1: Whether the algorithm is less practical***
>
> Thank you for the comments. Firstly, we would like to clarify that the limited practicality arises from the inherent hardness of the adversarial transitions problem, rather than from any weakness in our algorithmic design. Our minimax optimal regret bound provides a fundamental understanding and a new perspective on the fully adversarial transitions setting.
>
> Besides, our algorithm is computational economic when applying policy-optimization version (Please check the discussion with Reviewer sqJH and Appendix B.3). Therefore, our algorithm can be applied in real application undoubtedly. Moreover, the key difference between our work and previous adversarial transitions works is that out algorithm could output a policy which can approximate the optimal policy. However, previous works could not output this kind of policy even suboptimal policy within the Markov policy set in unknown fully-adversarial transitions.
>
> ***Weakness 2: Related work discussion***
>
> Thanks for your comments. We have updated the new version and refined the corresponding descriptions.
>
> ***Weakness 3: Discussion in Remark 1***
>
> Thanks for your comments. We have updated the new version and incorporated this case into the discussion.
>
> ***Q1: Deeper discussion of extra challenges due to adversarial reward functions***
>
> Thank you for the comments. Since adversarial and unknown transitions constitute the key challenge and main contribution of our work, we primarily focus on the difficulties arising from *adversarial transitions*. In fact, our AD-FTRL algorithm can also be applied to settings with *adversarial reward functions* under unknown adversarial transitions. The bandit feedback introduces additional challenges, particularly in estimating the full reward function.
>
> Moreover, compared with previous works, our AD-FTRL algorithm is based on the **trajectory occupancy measure** and can handle environments with **trajectory-wise reward functions** rather than only state–action-wise reward functions. This represents a key advantage of our approach over existing methods and provides substantial benefits for reward-based applications.
>
> ***Question 2: Results with varying state/action space sizes; reason why assume the action/state sets are non-overlapping***
>
> Thanks for the comments. If the sizes of the state and action sets differ across stages, the results become
> $\prod\_{k=0}^{K-1} |\mathcal{A}\_k|$ instead of $|\mathcal{A}|^K$,
> and $\prod\_{k=0}^{K} |\mathcal{S}\_k|$ instead of $|\mathcal{S}|^K$ (in our work, $|\mathcal{S}\_0| = 1$).
>
> Furthermore, we assume that the state and action sets are non-overlapping across different stages, since we consider finite-horizon non-stationary MDPs where the transition dynamics can vary from one stage to another within an episode. Consequently, even if some actions share the same name or physical meaning, their effects may differ across stages. Therefore, actions taken at different stages should not be regarded as identical. Following this perspective, we—consistent with many previous works—assume that the action sets are disjoint across stages within an episode. This assumption not only reflects the practical characteristics of such environments but also simplifies the formulation for better readability.
>
> Moreover, setting the state and action spaces to have the same dimensions across different stages does not reduce the technical challenges. Instead, this choice helps avoid a cumbersome proof process and improves readability.
>
> ***Question 3: Works in related research areas***
>
> Thanks for your comments. We have updated the new version and refined the corresponding descriptions.

---

> > ### Comment · Reviewer_jRLS · 2025-11-28
> >
> > Thanks to the authors for the great response! I keep my score for marginal accept.

---

### Official Review · Reviewer_sqJH · 2025-10-31

**Soundness:** 3
**Presentation:** 3
**Contribution:** 4
**Rating:** 8
**Confidence:** 4

**Summary:**

This work studies how to learn MDPs with both adversarially chosen transition kernels and loss functions. Prior studies only obtain a regret of $\mathcal{O}(\sqrt{T}+C^P)$, where $C^P$ is the degree of adversarial change of transition kernels. This work proposes an FTRL algorithm, which operates over the space of trajectory occupancy measure and achieves a regret of $\mathcal{O}\left(\sqrt{(|\mathcal{S}||\mathcal{A}|)^K T}\right)$. The authors also prove a matching lower bound for this problem.

**Strengths:**

1. **Novelty**: The idea of the trajectory occupancy measure in this work is simple yet powerful. The matching lower bound is also very interesting.
2. **Presentation**: Most parts of this work are generally well-written and easy to follow.

**Weaknesses:**

1. **Computation efficiency**: If any, I feel that operating over the space of the trajectory occupancy measure makes the algorithm computationally inefficient. Also, I seemingly cannot find any discussions relating to the computational efficiency of the proposed algorithm. That said, I personally do not think it is a significant drawback, considering that this work proposes the first minimax optimal algorithm for this challenging problem.

**Questions:**

1. In MDPs with unknown yet fixed transition kernels and adversarially chosen loss functions, we can devise a policy optimization (PO)-based algorithm, which is more computationally efficient and easier to implement. Is it possible to devise a similar PO-based algorithm for the problem considered in this work?
2. Typically, using FTRL with Shannon entropy may lead to suboptimal factors of $\log A$ in adversarial bandits and of $\log (SA)$ in adversarial RL. Why FTRL with Shannon entropy can lead to the minimax optimal regret without having similar factors in the result?

---

> ### Author Response · Authors · 2025-11-21
>
> ***Weaknesses: Discussion on computational efficiency***
>
> Thank you for the comments. The main computational bottleneck lies in *Line 12 of Algorithm 1*, where the computational cost arises from solving an optimization problem defined over a space of size $(|\mathcal{S}||\mathcal{A}|)^K$. This step dominates the overall computational cost of each iteration of the algorithm.
>
> Combined with the discussion in Q1, we have added a detailed analysis of the computational cost in Appendix B.3.
>
> ***Q1: Policy Optimization (PO)-based Algorithm***
> We appreciate the insightful comments. In addition to the main algorithm, we provide the algorithm based on the \textbf{mirror descent} framework in Appendix B. We agree that this PO-based formulation is more computationally efficient. Next, we provide the detailed analysis. The update rule in Algorithm 2 is given by
> $$\Pi\_{t+1} = \Pi\_{t=0}^{K-1} \pi\_{t+1}(a^k|h^k);\pi\_{t+1} = \arg\min\_{\pi \in \mathcal{C}\_{\pi, \epsilon}} \left( \langle \Pi\_\pi, \widehat{\mathcal{L}}\_{t} \rangle + \frac{1}{\eta\_{t+1}} \Phi\_{t+1}(\Pi\_\pi) \right),$$
> where $\ell\_{\tau\_t} = \sum\_{k=0}^{K} \ell\_t(s^k\_t, a^k\_t), \quad \widehat{\mathcal{L}}\_t = \frac{\ell\_{\tau\_t}}{\gamma + \prod\_{k=0}^{K-1} \pi\_t(a^k\_t \mid h^k\_t)} [\mathbb{I}\_{\tau=\tau\_t}]\_{\tau \in \mathcal{C}\_\tau}.$
>
> By applying the KKT conditions and noting that $\widehat{\mathcal{L}}\_{t}$ contains only one nonzero entry, we obtain the following update: $\pi\_{t+1}(a^k|h^k) \propto \pi\_t(a^k|h^k) \exp(-\eta\_t \widehat{\mathcal{L}}\_{t}).$
>
> Hence, for all $a, h, k$, the policy can be iteratively updated as:
> $$\pi\_{t+1}(a^k|h^k)= \frac{ \pi\_t(a^k|h^k)\exp(-\eta\_t \ell\_{\tau\_t} \mathbb{I}(\{h^k,a^k\} \in \tau\_t)) }
> {\sum\_{a\in \mathcal{A}^k}\pi\_t(a|h^k)\exp(-\eta\_t \ell\_{\tau\_t} \mathbb{I}(\{h^k,a\} \in \tau\_t))}.$$
>
> Under this formulation, the computational cost is reduced from $\mathcal{O}((|\mathcal{S}||\mathcal{A}|)^K)$ to $\mathcal{O}(|\mathcal{S}||\mathcal{A}|K)$.
> Therefore, the proposed *PO-based method* offers a more efficient alternative. We have added these part in the appendix.
>
> ***Q2: The Shannon entropy can lead to suboptimal factors***
>
> Thanks for the comments. The use of Shannon entropy indeed introduces an additional sub-optimal term of order $\mathcal{O}(K \log(|\mathcal{S}||\mathcal{A}|))$. However, this term is negligible compared to the dominant term $\mathcal{O}((|\mathcal{S}||\mathcal{A}|)^K)$. Therefore, we omit this factor in our analysis. In this work, we primarily focus on the dependence of the regret on the leading-order term $\mathcal{O}((|\mathcal{S}||\mathcal{A}|)^K).$

---

### Official Review · Reviewer_jyEJ · 2025-11-01

**Soundness:** 3
**Presentation:** 3
**Contribution:** 3
**Rating:** 6
**Confidence:** 3

**Summary:**

The paper studies episodic RL with fully adversarial transitions and bandit feedback. At every round the environment can pick an arbitrary MDP (transitions + losses) from a hard family; the learner only observes a trajectory. The authors propose a trajectory-level FTRL algorithm that maintains a distribution over trajectories and uses importance-weighted loss estimates. They prove a regret upper bound $O((|S||A|)^{\frac{K}{2}}\sqrt{T})$ and then match it with a lower bound of the same order. Thus, the minimax optimal rate is derived for this model. They also show Markov policies can be suboptimal: optimal strategies must be history-dependent.

**Strengths:**

The paper give sharp information-theoretical result. Upper and lower bounds match with order $(|S||A|)^{\frac{K}{2}}\sqrt{T}$ so the rate is tight for fully adversarial model.

By working directly on trajectory occupancies instead of state–action occupancies, the paper pinpoints where previous bounds picked up a linear term in the transition variation.

The example showing Markov policies are insufficient in this adversarial setting is useful and interesting.

**Weaknesses:**

Both the regret bound and computational complexity is in exponentially order. Although this is correct and tight in theory, it makes the algorithm less practical.

In previous papers where $O(\sqrt{T} + C^P)$ regret is derived, it gets sublinear regret when the change of transition is small. This paper, however, always suffer  $(|S||A|)^{\frac{K}{2}} \sqrt{T}$ regret no matter how the transition changes. Thus, the new algorithm is better when $C^P = O(T)$ and $T >  (|S||A|)^{K}$, which is a very restrictive regime.

The change of transition is obliviously adversarial, what is the additional challenge for adaptive adversary?

**Questions:**

See weakness.

---

> ### Author Response · Authors · 2025-11-21
>
> ***Q1: Whether the algorithm is less practical***
>
> Thank you for the comments. Firstly, we would like to clarify that the limited practicality arises from the inherent hardness of the adversarial transitions problem, rather than from any weakness in our algorithmic design. Our minimax optimal regret bound provides a fundamental understanding and a new perspective on the fully adversarial transitions setting.
>
> Besides, our algorithm is computational economic when applying policy-optimization version (Please check the discussion with Reviewer sqJH and Appendix B.3).
>
> Furthermore, we acknowledge the concern regarding the limited practicality caused by the exponential term. This issue also motivates an important direction for future research: building upon our current theoretical framework to investigate whether, under additional assumptions or constraints (e.g., constant transitions), a more practical algorithm can be developed.
>
> ***Q2: Whether the sublinear regret regime is strict***
>
> Firstly, we need to claim that the linear $\mathcal{O}(T)$ transitions corruption is a widely used situation. For example, if there exist two different transitions and these two transitions occur alternately, the transition corruption level is linear in $\mathcal{O}(T)$. These setting can not be cover to sublinear corruption setting, which usually has a nominal transitions and little corruption. Therefore, the linear $\mathcal{O}(T)$ transitions corruption is a significant and widely used situation.
>
> Secondly, the large-scale dates are widely used. Therefore, the situation that the iteration size achieves $\mathcal{O}((|\mathcal{S}||\mathcal{A}|)^K)$ is a general situation in large-scale date setting.
>
> Besides, the key difference between our work and previous adversarial transitions works is that out algorithm could output a policy which can approximate the optimal policy. However, previous works could not output this kind of policy even suboptimal policy within the Markov policy set in unknown fully-adversarial transitions.
>
> ***Q3): Extra challenges in the adaptive adversarial setting***
>
> Thank you for your valuable comments. Extending our work to the *adaptive adversarial* setting is non-trivial, as the fully adaptive adversarial MDP framework presents substantial challenges. Below, we discuss the implications for both the upper-bound and lower-bound analyses.
>
> **Upper bound**
> To the best of our knowledge, most existing algorithms in the adaptive adversarial literature consider the setting with *fixed transition kernels*. We believe that such methods can be extended to handle *adaptive adversarial transitions* by integrating our proposed framework.
>
> However, applying our analysis directly to the adaptive adversarial setting introduces several additional challenges.
> First, there exist two notions of regret in this setting:
> (1) the *static regret*, measured against a fixed optimal policy, and
> (2) the *dynamic regret*, measured against an optimal sequence of policies.
> Our current framework can be extended to analyze the static regret, but under adaptive adversaries, the value function depends on the entire episode history, which introduces significant analytical difficulty.  Moreover, the analysis of the memory-dependent components also becomes substantially more complex.  We believe that addressing these challenges represents an important and insightful direction for future work.
>
> **Lower bound**
> The extension of the lower-bound result is relatively straightforward. Our lower bound holds for **all algorithms**, and since the oblivious adversary is a weaker version of the adaptive adversary, the same regret lower bound (against a fixed policy) directly applies to the adaptive adversarial transition setting as well.

---

### Official Review · Reviewer_rRoE · 2025-11-06

**Soundness:** 3
**Presentation:** 3
**Contribution:** 3
**Rating:** 6
**Confidence:** 4

**Summary:**

This submission introduces (to the adversarial RL community) the key observation that the optimal policy in episodic MDPs with adversarial trantisions and rewards is not necessarily Markovian. Following this observation, the authors give a FTRL-type algorithm achieving the optimal regret $\tilde{O}(\sqrt{|\mathcal{S}|^K|\mathcal{A}|^K}\sqrt{T})$. The authors also certificate the optimality of the standard FTRL routine with a novel lower bound construction, simutaneously manifesting the exponential-in-$K$ dependency of $|\mathcal{S}|$ and $|\mathcal{A}|$.

**Strengths:**

- The employment of non-Markovian policy as the correct class to consider is a novel and significant observation
- The construction beyond constant many arms is novel for proving the lower bound
- Lemma B.3, which justify the convexity of the set of trajectory distributions induced by $\epsilon$-greedy non-Markovian policies, appears to be novel

**Weaknesses:**

- Algorithm 1 a relatively familiar to the adversarial RL community, if not standard.
  - It is intuitively likely that history-dependent policy has been considered in different contexts but also under the algorithmic framework of occupancy measure; with that said, I respectfully question the originality of Lemma B.3. The authors can correct me if that is not the case.
  - Since Lemma B.3 seems to be the major technical novelty in the upper bound analysis, it is more appropriate for the authors to justify its originality.


### Minor weaknesses

- On Figure 1: There is as a typo because the authors may want to design the initial state in $P_1$ to be $\star$.
- Theorem 5.1: $O$ -> $\tilde{O}$
- Line 320: The first $\pi$ on the RHS should be $\pi_t$
- Line 226: Occupancy measure has been introduced in very classical literature, e.g., those regarding the linear programming formulation of RL in MDPs. I mean this is definitely not a concept that was introduced for the first time in the 21st century.
- Line 169: It seems that the loss function considered within one episode is time-homegeneous, right?

**Questions:**

- Given the existence of EXP3.P in the literature that achieves high-probability guarantees, do the authors plan to argue about the dfficulty for proving a high-probability regret bound for (potentially certain variant) of their Algorithm 1?
- The lower bound construction is more significant than the upper bound analysis. In fact, is it more appropriate to reduce the length of the description of the relatively standard FTRL-type routines in the main text and elaborate more on how the authors successfully manifest the dependency on $|\mathcal{S}|^{\Omega(K)}$ and $|\mathcal{A}|^{\Omega(K)}$ simultaneously in the lower bound analysis?
- Though it is relatively standard for adversarial RL papers to assume the state space to be disjoint for different steps, it is a bit interesting to also assume the action set to be disjoint for different steps. How does this later assumption simplifes the proof?

---

> ### Author Response · Authors · 2025-11-21
>
> ***Weaknesses: the originality of Lemma B.3***
>
> Thank you for your comments.
>
> First, we would like to clarify that our work focuses on *trajectory occupancy measures*, rather than *state--action occupancy measures*. Prior works (Rosenberg \& Mansour, 2019; Jin et al., 2020a,b) have established that the set of state--action occupancy measures induced by Markov policies is convex. In contrast, Lemma B.3 in our paper proves that the set of trajectory occupancy measures is convex when the policy class consists of history-dependent policies. Notably, the trajectory occupancy measures set in our work is **not convex** when restricted to **Markov** policies.
>
> We understand the reviewer's concern. However, after reviewing the related literature, we could not find any prior results that establish the convexity of trajectory occupancy measures under history-dependent policy classes.
>
> Finally, we emphasize that the main technical novelty of our work lies in the *construction of the hard-MDP instances* for the lower bound and the *proof of the minimax regret bound*. For the upper bound, our contribution centers on *algorithm design*, where we propose a practical method for handling unknown adversarial transition kernels under bandit feedback.
>
> ***Minor weaknesses:***
>
> Thanks for the comment. We have updated the new version for the corrected information.
>
> Line 226: We have updated the description and polished it to make it more professional.
>
> Line 169: The loss functions are assumed to be time-homogeneous within each episode, but time-varying or adversarial across different episodes.
>
> ***Question 1: Providing a high-probability regret bound.***
>
> Thank you for your comment. In our work, we provide the expectation-based version of the regret bound rather than the high-probability version. However, the high-probability result can also be derived from the proof (see Appendix, Eq. (12)). We have now added a new theorem (Theorem B.2) in the Appendix that explicitly presents the *high-probability* version of the regret bound.
>
> ***Question 2: Lower bound as the major contribution. Provide a deeper discussion of the lower bound.***
>
> First, we thank the reviewer for recognizing the technical improvement and novelty of our lower-bound analysis. The design of the lower bound constitutes a significant technical contribution of our work. To the best of our knowledge, this is the first work to formally structure and present this proof framework. Moreover, our approach can be extended to other related research areas.
>
> Specifically, regarding the dependence on the $\Omega(|\mathcal{S}|^{K/2})$ term, we can reduce the problem to a contextual multi-armed bandit with $|\mathcal{S}|^K$ contexts and $|\mathcal{A}|$ arms—each corresponding to a possible history and action in the last stage—yielding a regret lower bound of order $\Omega(\sqrt{|\mathcal{S}|^K |\mathcal{A}| T})$. However, establishing the dependence on the $\Omega(|\mathcal{A}|^{K/2})$ term is more challenging.  To illustrate the additional difficulty, consider the construction of the hard MDPs. In the first stage, starting from the initial state $s_0$, it is necessary to identify the optimal action. The same challenge persists in subsequent stages: if the policy chooses a suboptimal action, the observed feedback remains indistinguishable from that of the optimal one. Without prior knowledge of the optimal policy in later stages, at stage $k$, random exploration leads to an average feedback gap between optimal and suboptimal actions of approximately $\frac{\epsilon}{|\mathcal{A}|^{K-k-1}}$. Furthermore, there are $|\mathcal{S}|^{k}$ optimal actions that must be identified. This provides an intuitive explanation for the $\Omega(|\mathcal{A}|^{K/2})$ dependence.  A rigorous mathematical justification of this result relies on the use of the \textit{composite hypothesis testing theorem} to establish the corresponding lower bound.
>
> We are working on a detailed proof sketch and will update it when finished.
>
> ***Question 3: Why is the action set to be disjoint for different steps?***
>
> Thank you for the comment. In our work, we assume that the finite-horizon MDPs are non-stationary, meaning that the transition dynamics vary across different stages within an episode. Therefore, even if some actions share the same name or physical meaning, their effects differ at different stages. As a result, actions taken at different stages should not be regarded as the same. Following this perspective, we (as well as many previous works) assume that the action sets are disjoint across different stages within an episode. This setting reflects the practical characteristics of such environments and also makes the formulation easier for readers to understand. Moreover, it does not reduce the technical challenges but helps avoid a cumbersome proof process and improves readability.

---

> > ### Comment · Reviewer_rRoE · 2025-11-21
> >
> > Thank you for the detailed response. I believe I have understood and appreciated the significance of this submission, especially the constructive proof of the hardness results for non-Markovian policies. I keep my positive evaluation of this submission unchanged, which votes the acceptance.

---

### Meta-Review · Area_Chair_3t5N · 2025-12-30

**Summary:**

The reviewers broadly agree that this paper makes a strong theoretical contribution to adversarial reinforcement learning by addressing the fully adversarial transition setting with bandit feedback and establishing minimax-optimal regret bounds. They recognize the importance of the insight that optimal policies must be history-dependent and view the matching upper and lower bounds as the main strength of the work. Concerns raised by reviewers mainly relate to limited practical applicability due to an exponential dependence on problem parameters and to the clarity of the work's positioning relative to prior adversarial and non-stationary RL literature. No reviewer questioned the soundness or correctness of the results, and overall, the paper received consistent marginally positive evaluations.

**Reviewer Concerns:**

The rebuttal addressed several key concerns. In particular, it clarified the originality of the trajectory-occupancy measure analysis and the convexity result for history-dependent policies, and it strengthened the discussion of computational complexity by acknowledging inherent hardness and presenting a more efficient policy-optimization variant. The authors also improved the positioning of the work relative to prior literature, clarified modeling assumptions, and addressed questions about high-probability guarantees and adaptive adversaries. Some concerns about limited practicality and restrictive regimes remain, but these are inherent primarily to the fully adversarial setting and do not detract from the paper’s theoretical contributions.

**Reviewer Scores:**

Based on the rebuttal and existing discussion, one reviewer would likely have increased their score after concerns about originality were resolved, while the remaining reviewers would likely have maintained their original scores. Overall, the inferred post-rebuttal consensus remains slightly positive, supporting acceptance.

---

### Decision · Program_Chairs · 2026-01-26

Accept (Poster)